# Parkin contributes to synaptic vesicle autophagy in Bassoon-deficient mice

Sheila Hoffmann-Conaway[1], Marisa M Brockmann[2,3], Katharina Schneider[1], Anil Annamneedi[4,5,6], Kazi Atikur Rahman[1,7], Christine Bruns[1], Kathrin Textoris-Taube[8], Thorsten Trimbuch[2,3], Karl-Heinz Smalla[4,5], Christian Rosenmund[2,3], Eckart D Gundelfinger[4,5,9], Craig Curtis Garner[1,2,3]*, Carolina Montenegro-Venegas[4,5,10]*

[1]German Center for Neurodegenerative Diseases (DZNE), Berlin, Germany; [2]Charité – Universitätsmedizin Berlin, Institute of Neurobiology, Berlin, Germany; [3]NeuroCure Cluster of Excellence, Charité – Universitätsmedizin Berlin, Berlin, Germany; [4]Leibniz Institute for Neurobiology, Magdeburg, Germany; [5]Center for Behavioral Brain Sciences (CBBS), Magdeburg, Germany; [6]Institute of Biology (IBIO), Otto von Guericke University Magdeburg, Magdeburg, Germany; [7]Einstein Center for Neurosciences Berlin, Berlin, Germany; [8]Charité – Universitätsmedizin Berlin, Institute of Biochemistry, Core Facility High Throughput Mass Spectrometry, Berlin, Germany; [9]Molecular Neurobiology, Medical Faculty, Otto von Guericke University, Magdeburg, Germany; [10]Institute for Pharmacology and Toxicology, Medical Faculty, Otto von Guericke University, Magdeburg, Germany

*For correspondence:
craig.garner@dzne.de (CCG);
cmontene@lin-magdeburg.de (CM-V)

Competing interests: The authors declare that no competing interests exist.

**Abstract** Mechanisms regulating the turnover of synaptic vesicle (SV) proteins are not well understood. They are thought to require poly-ubiquitination and degradation through proteasome, endo-lysosomal or autophagy-related pathways. Bassoon was shown to negatively regulate presynaptic autophagy in part by scaffolding Atg5. Here, we show that increased autophagy in *Bassoon* knockout neurons depends on poly-ubiquitination and that the loss of Bassoon leads to elevated levels of ubiquitinated synaptic proteins per se. Our data show that *Bassoon* knockout neurons have a smaller SV pool size and a higher turnover rate as indicated by a younger pool of SV2. The E3 ligase Parkin is required for increased autophagy in *Bassoon*-deficient neurons as the knockdown of *Parkin* normalized autophagy and SV protein levels and rescued impaired SV recycling. These data indicate that Bassoon is a key regulator of SV proteostasis and that Parkin is a key E3 ligase in the autophagy-mediated clearance of SV proteins.

## Introduction

Presynaptic active zones, the sites of synaptic vesicle fusion, are comprised of a dense network of scaffold proteins. These function to coordinate the regulated release of neurotransmitters as well as the cellular machinery that maintains the function and integrity of presynaptic boutons and thus neuronal communication (*Rangaraju et al., 2014*; *Wang et al., 2017*). The post-mitotic nature of neurons and their long lifetimes likely contributes to their vulnerability to various kinds of stress (*Bishop et al., 2010*) often culminating in the accumulation of misfolded proteins and neuronal degeneration as clearance systems become exhausted. Recent studies point to the presence of several proteostatic mechanisms within or near synapses that function to remove non-functional and/or damaged proteins. These include the proteasome, the endo-lysosomal system and autophagy (*Hoffmann et al., 2019*; *Kaushik and Cuervo, 2015*; *Lazarevic et al., 2011*; *Vijayan and Verstreken, 2017*; *Wang et al., 2017*). Genetic or environmental insults to these clearance systems

cannot only disrupt synaptic transmission but also trigger developmental and neurodegenerative disorders (*Liang and Sigrist, 2018*; *Vijayan and Verstreken, 2017*; *Waites et al., 2013*).

Recently, it has been shown that synaptic vesicle proteins can enter the autophagy pathway, e.g. after being damaged by reactive oxygen species (ROS) (*Hoffmann et al., 2019*). However, how exactly these proteins are labeled for degradation remains unknown. One tagging system involved in protein quality control is the ubiquitination system, which attaches specific ubiquitin chains to a substrate, potentially leading to its degradation through either the proteasome or the lysosome. In addition, recent evidence suggests that ubiquitination plays an essential role in the generation of autophagic structures (*Ciechanover and Kwon, 2017*; *Deng et al., 2017*).

Ubiquitination requires the regulated activation of several components of this tagging system, including the ubiquitin activating enzyme E1, E2 ubiquitin conjugating enzymes and E3 ubiquitin ligases (E3 ligases) (*Ding and Shen, 2008*; *Nandi et al., 2006*). E3 ubiquitin ligases, of which mammalian cells express several hundred (*Li et al., 2008*), are posited to be responsible for substrate and degradation pathway specificity (*Kwon and Ciechanover, 2017*). For example, K63-poly-ubiquitinated proteins can be recognized by the autophagy adaptor p62, leading to the engulfment of these proteins into autophagy organelles and their subsequent delivery to the lysosome (*Linares et al., 2013*). On the other hand, K48-poly-ubiquitinated proteins are generally recruited into the proteasome for degradation (*Ji and Kwon, 2017*).

Interestingly, at least two key regulators of synaptic transmission, RIM1 and Munc13, can be selectively removed by the proteasome in an E3 ubiquitin ligase-dependent manner (*Jiang et al., 2010*; *Yao et al., 2007*; *Yi and Ehlers, 2005*). Additionally, the knockdown of two multi-domain scaffold proteins at the active zone (AZ), Piccolo and Bassoon, led to the loss of SVs and overall synapse integrity (*Okerlund et al., 2017*; *Waites et al., 2013*). This dramatic effect was caused in part by the activation of the E3 ubiquitin ligase Siah1 (*Waites et al., 2013*). Intriguingly, hippocampal neurons from mice lacking Bassoon alone show a robust increase in autophagosomes throughout axons and within synapses (*Okerlund et al., 2017*). These findings suggest that Bassoon is a potent regulator of presynaptic proteostasis, to some extent through the scaffolding constituents of the autophagy pathway such as Atg5 (*Okerlund et al., 2017*).

At present, it is unclear which E3 ligases are critical for enhanced presynaptic autophagy observed in the absence of Bassoon as well as which presynaptic proteins ultimately become degraded when the autophagy system is activated. In this study, we sought to address several questions. Specifically, what is the cargo of autophagy induced by the lack of Bassoon? Is the ubiquitin system involved in the substrate tagging and delivery in *Bassoon* knockout (*Bsn* KO) mice? And which E3 ligases are required to drive Bassoon deficiency-triggered autophagy? Regarding the latter, we focused our studies on two ligases, the RING-type E3 ligase Siah1 and the RING-between-RING-type (RBR-type) E3 ligase Parkin, as both are known to ubiquitinate several SV-associated and other presynaptic proteins (*Chung et al., 2001*; *Trempe et al., 2009*; *Wheeler et al., 2002*). Parkin is particularly attractive, given its association with several neurodegenerative diseases including Parkinson's disease (*Corti et al., 2011*; *Nixon, 2013*). Moreover, Parkin is well known for being involved in mitophagy, the autophagic clearance of dysfunctional mitochondria (*Ashrafi et al., 2014*; *Pickrell and Youle, 2015*) and to generate K63-poly-ubiquitinated substrates (*Olzmann et al., 2007*) that can be degraded by autophagy (*Linares et al., 2013*). Our analysis reveals that Bassoon deficiency-triggered presynaptic autophagy requires the ubiquitin system, which primarily ubiquitinates SV proteins through the activation of Parkin and to a lesser extent Siah1. Overactivation of this system has a negative impact on SV recycling and pool size, suggesting a novel role for the E3 ubiquitin ligase Parkin in SV clearance.

## Results

### Increased autophagy in *Bassoon* knockout neurons depends on poly-ubiquitination

The primary goal of this study was to understand how presynaptic boutons regulate the catabolism of SV proteins as they age. Our previous study identified the presynaptic active zone proteins Bassoon and Piccolo as key regulators of SV proteostasis as well as synaptic integrity (*Okerlund et al., 2017*; *Waites et al., 2013*). We could also show that Bassoon was involved in the local regulation of

autophagy within presynaptic boutons in part by scaffolding the autophagy protein Atg5 (*Okerlund et al., 2017*), and that ROS-mediated damage of SV proteins can trigger their clearance via autophagy (*Hoffmann et al., 2019*). Conceptually, presynaptic autophagy could be trigged by the generation of ubiquitinated substrates (*Kwon and Ciechanover, 2017*), or via the activation of signaling systems that e.g. inactivate mammalian target of rapamycin (mTOR) (*Codogno et al., 2012*; *Hernandez et al., 2012*; *Klionsky et al., 2012*). At present, it is unclear whether Bassoon deficiency-induced autophagy is controlled by either or both. If the latter, one might anticipate that Bassoon loss of function only creates empty phagophore membranes, while the former should also trigger the creation of ubiquitinated substrates. Clearly the identification of such substrates could help to unravel the cellular program regulated by Bassoon, for instance whether it controls the catabolism of SV or other synaptic proteins.

Given that the N-terminal zinc fingers of Bassoon are known to bind and inhibit the RING-type E3 ubiquitin ligase Siah1 (*Waites et al., 2013*), we designed a series of experiments to test the hypothesis that Bassoon loss of function promotes autophagy by increasing the production of ubiquitinated substrates via specific E3 ligases. As an initial test of this hypothesis, we explored whether the expression of the two N-terminal zinc fingers (ZnFs) of Bassoon could suppress autophagy in neurons lacking Bassoon. This was accomplished by lentivirally expressing a construct with the first 609 amino acids of the Bassoon protein including ZnF1 and ZnF2 (Bsn609) tagged with eGFP in cultured hippocampal neurons from both WT and *Bsn* KO mice. As reported previously (*Dresbach et al., 2003*), Bsn609-eGFP properly localizes to the synapse (*Figure 1B*, *Figure 1—figure supplement 1A*). Co-transduction with RFP-LC3 in this and following experiments allowed for the monitoring of autophagy and a staining with antibodies against Synaptophysin1 was used to label synapses. In *Bsn* KO neurons, the introduction of FU-Bsn609-eGFP reduces the number of RFP-LC3 puncta per axon unit length back to WT levels (*Figure 1A,B and C*). Strikingly, the number of Synaptophysin1 puncta was significantly increased when FU-Bsn609-eGFP was expressed in *Bsn* KO neurons (*Figure 1A,B and D*). These findings are in line with *Waites et al., 2013*, showing that the expression of the Bassoon ZnFs could increase the intensity of SV2-eGFP at synapses in neurons expressing either shRNAs against both Piccolo and Bassoon or scrambled shRNA controls (*Waites et al., 2013*).

To further strengthen our hypothesis that Bassoon is acting locally to control autophagy, we performed western blot experiments showing that LC3-II levels are significantly elevated in synaptosomal preparations from *Bsn* KO mice (*Figure 1—figure supplement 2B and D*), but not in total brain homogenates (*Figure 1—figure supplement 2A and C*). As LC3-II is conjugated to the autophagosome (*Fleming et al., 2011*), it is an excellent marker for autophagy detection as the slight shift in size can be easily detected in western blot experiments.

To explore a more direct role of enhanced ubiquitination driving autophagy in boutons lacking Bassoon, we expressed a recombinant ubiquitin, with all lysine residues involved in poly-ubiquitination mutated to arginine ($UbK_0$), thus blocking the poly-ubiquitination of substrates (*Okerlund et al., 2017*; *Waites et al., 2013*), in hippocampal neurons from WT and *Bsn* KO mice. Although the endogenous ubiquitin is still present, this manipulation was sufficient to reduce the number of RFP-LC3 puncta (*Figure 2A,B and C*) and the colocalization of RFP-LC3 with Synaptophysin1 (*Figure 2A, B and E*) in *Bsn* KO neurons back to WT levels. Furthermore, we observed that inhibiting the E1 activating enzyme, by treatment of *Bsn* KO neurons with ziram (*Chou et al., 2008*; *Rinetti and Schweizer, 2010*), also reduces presynaptic autophagy levels back to control levels (*Figure 2—figure supplement 1C*, *Figure 1—figure supplement 1C*). These findings make a strong case that Bassoon may normally function to negatively regulate synaptic autophagy through its ability to inhibit enzymes involved in ubiquitination and/or poly-ubiquitination such as Siah1 (*Waites et al., 2013*).

## Increased levels of ubiquitinated SV proteins in *Bassoon* knockout neurons

To directly examine whether Bassoon deficiency enhances ubiquitination, we performed mass spectrometry analyses on synaptosomal preparations from WT and *Bsn* KO cortices. Using antibodies against the ubiquitin remnant motif (K-ε-GG) (*Udeshi et al., 2013*), we immunoprecipitated trypsin digested ubiquitinated peptides from synaptic proteins from synaptosomes. A scatterplot of WT and *Bsn* KO samples revealed a general shift towards more ubiquitinated synaptic proteins, such as SV-associated and SNARE complex proteins, in *Bsn* KO synaptosomes (*Figure 3*), indicating that Bassoon indeed negatively regulates ubiquitination of presynaptic proteins. Importantly, the

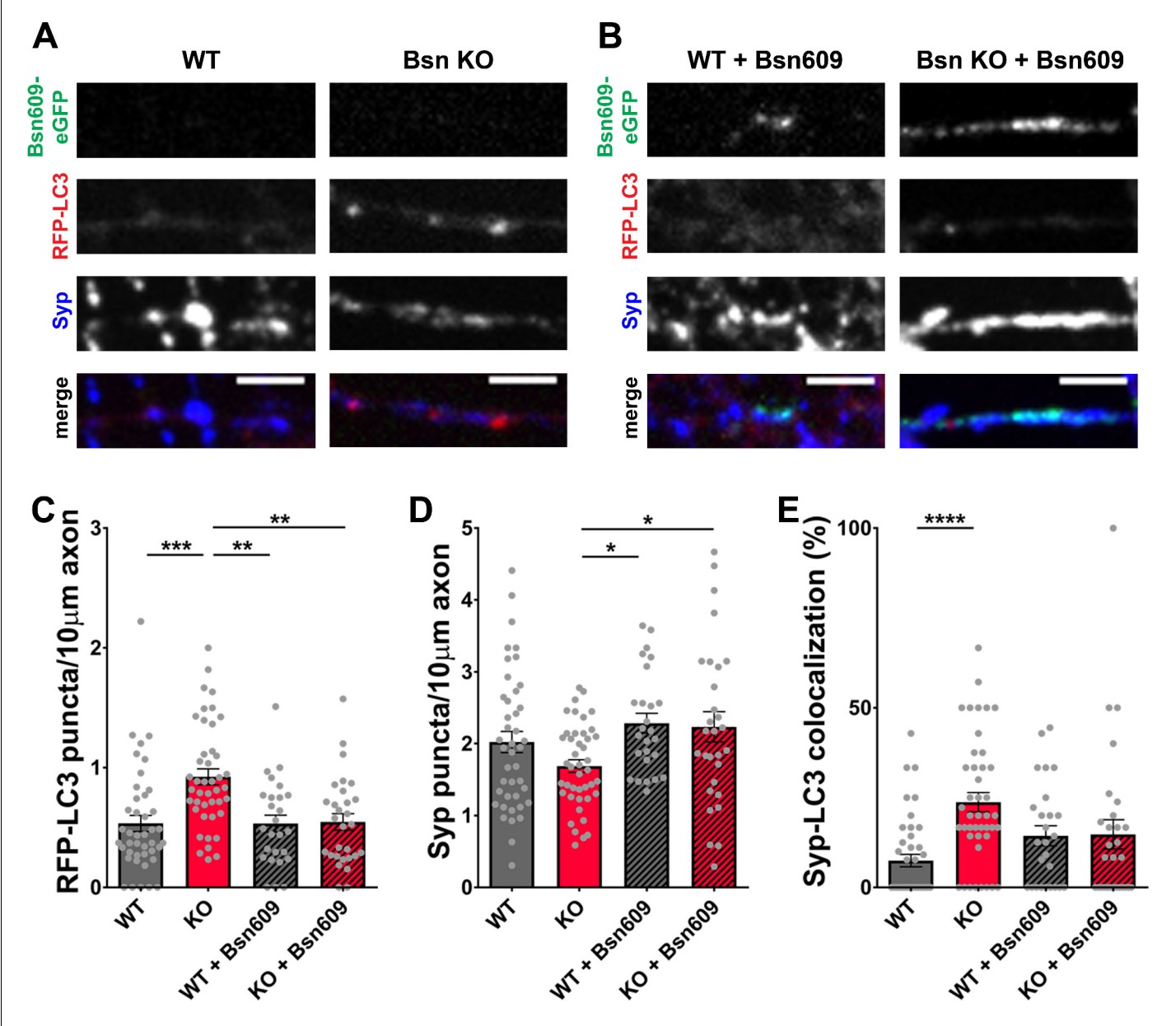

**Figure 1.** Expression of the Bsn fragment 1–609 (Bsn609) spanning both ZnF domains is sufficient to block autophagy induction in *Bassoon* KO neurons. (A–B) Representative images of hippocampal neurons from WT and *Bsn* KO mice expressing FU-RFP-LC3 only (A) or FU-RFP-LC3 and FU-Bsn609-eGFP (B) that were fixed and stained with antibodies against Synaptophysin1 (Syp). (C–E) Quantification of the number of RFP-LC3 puncta per 10 μm axon (C) (WT = 0.53 ± 0.067, n = 42 axons, 3 independent experiments; Bsn KO = 0.92 ± 0.066, n = 43 axons, 3 independent experiments; WT + Bsn609 = 0.53 ± 0.071, n = 26 axons, 3 independent experiments; Bsn KO + Bsn609 = 0.55 ± 0.069, n = 29 axons, 3 independent experiments; p***=0.0001, p**=0,001, p**=0.0011), the number of Syp1 puncta per 10 μm axon (D) (WT = 2.02 ± 0.148, n = 42 axons, 3 independent experiments; Bsn KO = 1.69 ± 0.088, n = 43 axons, 3 independent experiments; WT + Bsn609 = 2.28 ± 0.138, n = 26 axons, 3 independent experiments; Bsn KO + Bsn609 = 2.24 ± 0.210, n = 29 axons, 3 independent experiments; p*=0.0292, p*=0.0431) and the colocalization of RFP-LC3 and Syp1 (E) (WT = 7.45 ± 1.724, n = 42 axons, 3 independent experiments; Bsn KO = 23.71 ± 2.679, n = 43 axons, 3 independent experiments; WT + Bsn609 = 14.35 ± 2.796, n = 26 axons, 3 independent experiments; Bsn KO + Bsn609 = 14.74 ± 4.095, n = 29 axons, 3 independent experiments; p****<0.0001). Scale bars: 5 μm. Error bars represent SEM. Data points represent axons. ANOVA Tukey's multiple comparisons test was used to evaluate statistical significance.

The online version of this article includes the following figure supplement(s) for figure 1:

**Figure supplement 1.** Line scans of example images to illustrate colocalization of RFP-LC3 and Synaptophysin1.

**Figure supplement 2.** LC3-II levels are specifically increased in synaptosomal preparations from *Bassoon* KO mice.

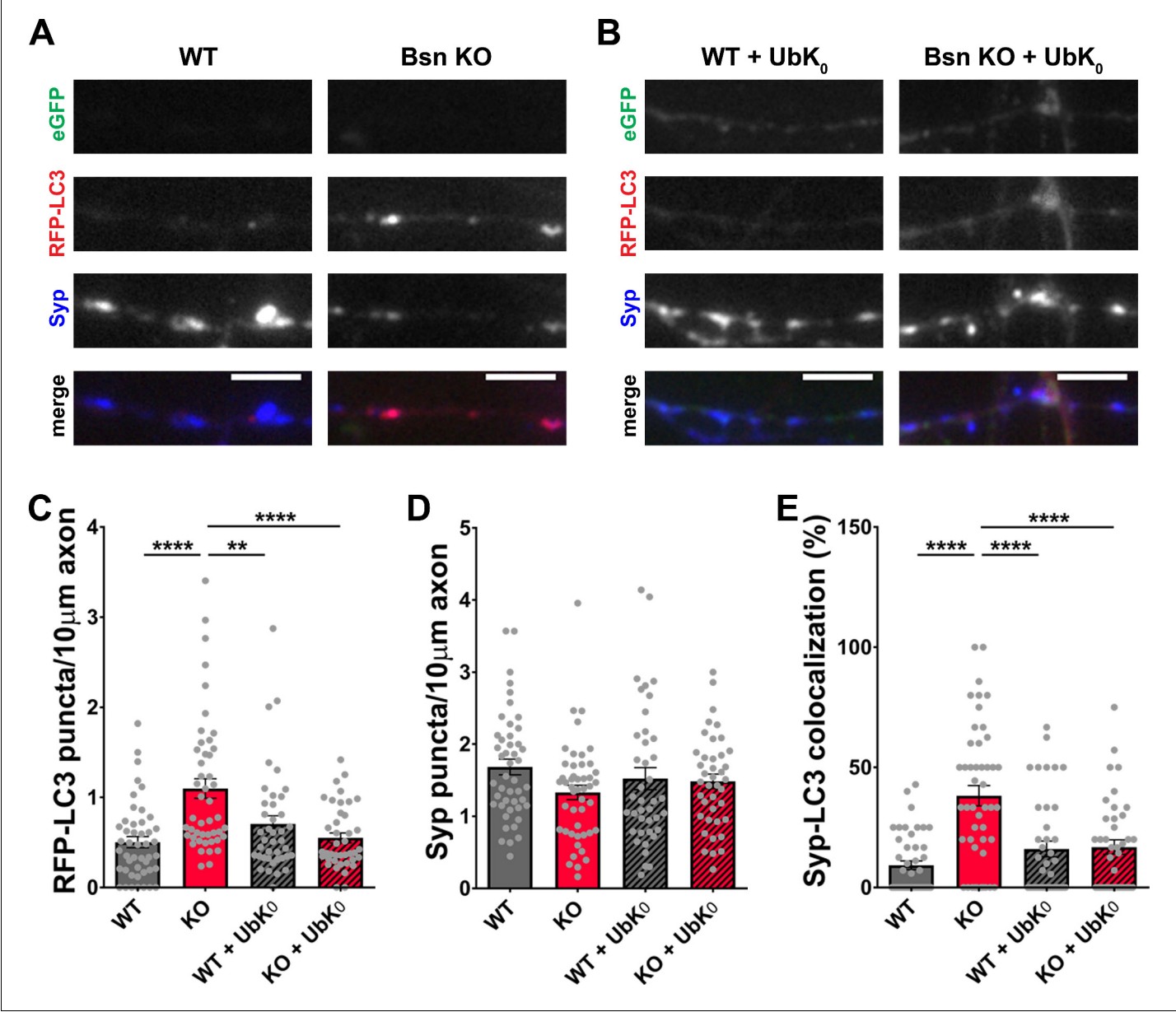

**Figure 2.** Poly-ubiquitination is required for increased autophagy in *Bassoon* KO neurons. (A–B) Representative images of hippocampal neurons from WT and *Bsn* KO mice expressing FU-RFP-LC3 only (A) or FU-RFP-LC3 and FU-UbK$_0$ (B) that were fixed and stained with antibodies against Synaptophysin1 (Syp). FU-UbK$_0$ vector co-expresses a soluble eGFP that enables the identification and quantification of axons from infected neurons only. (C–E) Quantification of the number of RFP-LC3 puncta per 10 μm axon (C) (WT = 0.50 ± 0.063, n = 46 axons, 3 independent experiments; Bsn KO = 1.10 ± 0.108, n = 47 axons, 3 independent experiments; WT + UbK$_0$ = 0.70 ± 0.091, n = 40 axons, 3 independent experiments; Bsn KO + UbK$_0$ = 0.55 ± 0.055, n = 40 axons, 3 independent experiments; p****<0.0001, p**=0.0060, p****<0.0001), the number of Syp1 puncta per 10 μm axon (D) (WT = 1.68 ± 0.109, n = 46 axons, 3 independent experiments; Bsn KO = 1.33 ± 0.100, n = 47 axons, 3 independent experiments; WT + UbK$_0$ = 1.52 ± 0.153, n = 40 axons, 3 independent experiments; Bsn KO + UbK$_0$ = 1.48 ± 0.102, n = 40 axons, 3 independent experiments) and the colocalization of RFP-LC3 and Syp1 (E) (WT = 9.23 ± 1.809, n = 46 axons, 3 independent experiments; Bsn KO = 38.19 ± 4.211, n = 47 axons, 3 independent experiments; WT + UbK$_0$ = 16.04 ± 3.269, n = 40 axons, 3 independent experiments; Bsn KO + UbK$_0$ = 16.82 ± 3.014, n = 40 axons, 3 independent experiments; p****<0.0001, p****<0.0001, p****<0.0001). Scale bars: 5 μm. Error bars represent SEM. Data points represent axons. ANOVA Tukey's multiple comparisons test was used to evaluate statistical significance.

The online version of this article includes the following figure supplement(s) for figure 2:

**Figure supplement 1.** Inhibition of E1 enzyme by ziram blocks autophagy induction in *Bassoon* KO neurons.

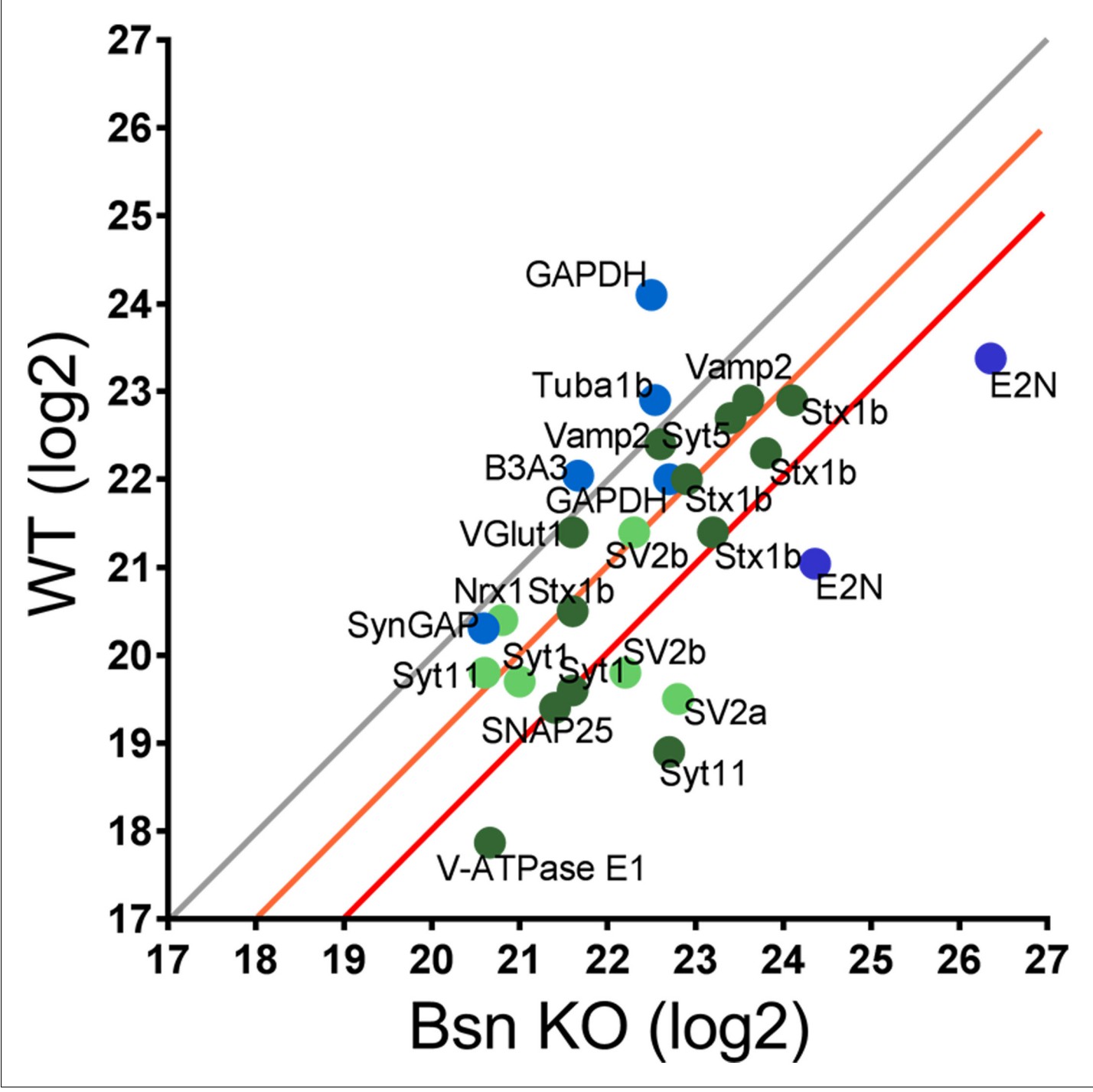

**Figure 3.** Mass spectrometry analyses reveal high levels of ubiquitinated SV-associated proteins in synaptosomal preparations from *Bassoon* KO mice. Before LC-MS/MS measurements, ubiquitinated peptides were immunoprecipitated with ubiquitin remnant motif antibody K-ε-GG. Scatterplot showing log2 intensities of detected ubiquitinated peptides (enriched for remnant GlyGly (K) sites) from synaptosomes prepared from cortices of 15-week-old WT and *Bsn* KO mice. Lysine-containing peptides can be ubiquitinated in more than one position (e.g. 5x Stx1b). Log2 intensities of IP enriched peptides from ubiquitously expressed and postsynaptic proteins in blue and presynaptic proteins in green (dark green = unambiguously identified, light green = ambiguously identified). Unambiguous identification equals Mascot score >40, $p_{pep}$ <0.001, ambiguous identification equals Mascot score <40 and $p_{pep}$ >0.001. The orange line describes a 2-fold and the red line a 4-fold increase. Data points represent ubiquitinated peptides. Three animals were used per genotype.

The online version of this article includes the following source data for figure 3:

**Source data 1.** Mass spectrometry analyses.

postsynaptic protein SynGAP and the ubiquitously expressed protein GAPDH were not altered in synaptosomes from *Bsn* KO mice, supporting a selective presynaptic change in ubiquitination.

Intriguingly, a variety of ubiquitinated presynaptic peptides are increased at least 4-fold in *Bsn* KO compared to WT neurons (*Figure 3*, *Figure 3—source data 1*): $SNAP25_{76}$, $Syt1_{119}$, $SV2b_{333}$, $SV2a_{143}$, $Syt11_{73}$, V-ATPase $E1_{10}$. A second set with a ~ 2 fold increase in ubiquitination (e.g. $Vamp2_{39,52,54}$, $Stx1b_{69}$, $Stx1b_{188}$, $Stx1b_{71}$, $Stx1b_{93}$, $Syt11_{124}$, $SV2b_{341}$, $Stx1b_{55}$, $Syt5_{63}$, $Syt1_{133}$) suggests that they may also be candidates for Bassoon-regulated ubiquitination. In experiments presented later, we were particularly drawn to SV2b and Vamp2 for further analysis of Bassoon-regulated SV turnover, as peptides from these SV proteins were increased in *Bsn* KO synaptosomes (*Figure 3*).

A further standout is E2N, for which we found ubiquitinated peptides that were more than 4-fold increased in *Bsn* KO synaptosomes (*Figure 3*). E2N is a ubiquitin conjugating enzyme directly involved in creating K63-poly-ubiquitin chains, known to tag proteins for degradation via autophagy (*Linares et al., 2013*). Increased levels of E2N ubiquitination would be consistent with its normal, yet higher, functional activity. Together these data support the concept that Bassoon loss of function drives the poly-ubiquitination of SV proteins down this clearance pathway.

## SV2b and Vamp2 levels as well as SV cycling are reduced in *Bassoon* knockout neurons and can be restored by the inhibition of autophagy

As the loss of Bassoon leads to increased ubiquitination levels of presynaptic proteins (*Figure 3*) while simultaneously inducing autophagy in the presynaptic compartment and throughout the axon, we anticipated that autophagy organelles could reduce the synaptic levels of these SV proteins by engulfing and delivering them to lysosomes for destruction. To test this concept, we quantified the synaptic levels of two SV proteins, SV2b and Vamp2, exhibiting increased ubiquitination in *Bsn* KO synaptosomes (*Figure 3*). Indeed, both SV2b and Vamp2 levels were reduced in *Bsn* KO neurons compared to WT neurons (*Figure 4A,D,B and E*). This effect was abolished by treating cultures with 1 µM of the autophagy inhibitor wortmannin (*Klionsky et al., 2012*; *Ravikumar et al., 2010*; *Figure 4A,D,B and E*), indicating that the loss of SV2b and Vamp2 in *Bsn*-deficient neurons is autophagy-dependent. To further strengthen these results, we performed similar experiments with SAR405, a highly specific Vps34 inhibitor (*Ronan et al., 2014*), to block autophagy. As expected, the treatment prevented autophagy induction as well as the decrease of SV2b levels in *Bsn* KO neurons (*Figure 4—figure supplement 1*).

The reduction in SV proteins such as SV2b and Vamp2 in *Bsn* KO neurons could be either caused by their specific elimination through autophagy (and other degradative pathways) or by the whole-sale removal of SVs. Both concepts could lead to an altered vesicle cycle function. Given that fluorophore-conjugated Synaptotagmin1 antibodies label SVs that underwent an exo-/endocytosis cycle (*Hoopmann et al., 2010*; *Truckenbrodt et al., 2018*), the amount of Synaptotagmin1 antibody uptake (Syt1 uptake) can be used as a measure of vesicle cycle efficiency under the assumption that the total amount of Synaptotagmin1 is unchanged. In a control experiment, there were no overt changes in Synaptotagmin1 levels in *Bsn* KO compared to WT neurons (*Figure 4—figure supplement 2A and B*) indicating that Syt1 uptake can be used as a measure of SV cycling. Interestingly, in *Bsn* KO neurons, we observed that the amount of activity-associated Syt1 uptake was significantly reduced. This could be rescued to WT levels by treating cultures with wortmannin, indicating that the reduced recycling of SVs in *Bsn* KO neurons is associated with increased autophagy (*Figure 4C and F*). Strikingly, the treatment with wortmannin alone in the WT background leads to a slight increase in Syt1 uptake (*Figure 4C and F*). This suggests that the acute inhibition of autophagy can increase SV cycles by itself, though the mechanism is unclear.

Together these findings imply that the increased ubiquitination of SV proteins, and in particular SV2b and Vamp2, in *Bsn* KO neurons reduces their levels through autophagy-mediated clearance. Conceptually, this could also lead to a decrease in the total uptake of Synaptotagmin1 antibody as there are fewer SVs to undergo recycling. However, as we also found enhanced ubiquitination of peptides from the t-SNARES, SNAP25 and Syntaxin1b, and the SV calcium sensor Synaptotagmin1 (*Figure 3*), the reduced exo-/endocytosis of SVs could be due to lower levels of these critical SV fusion-related molecules.

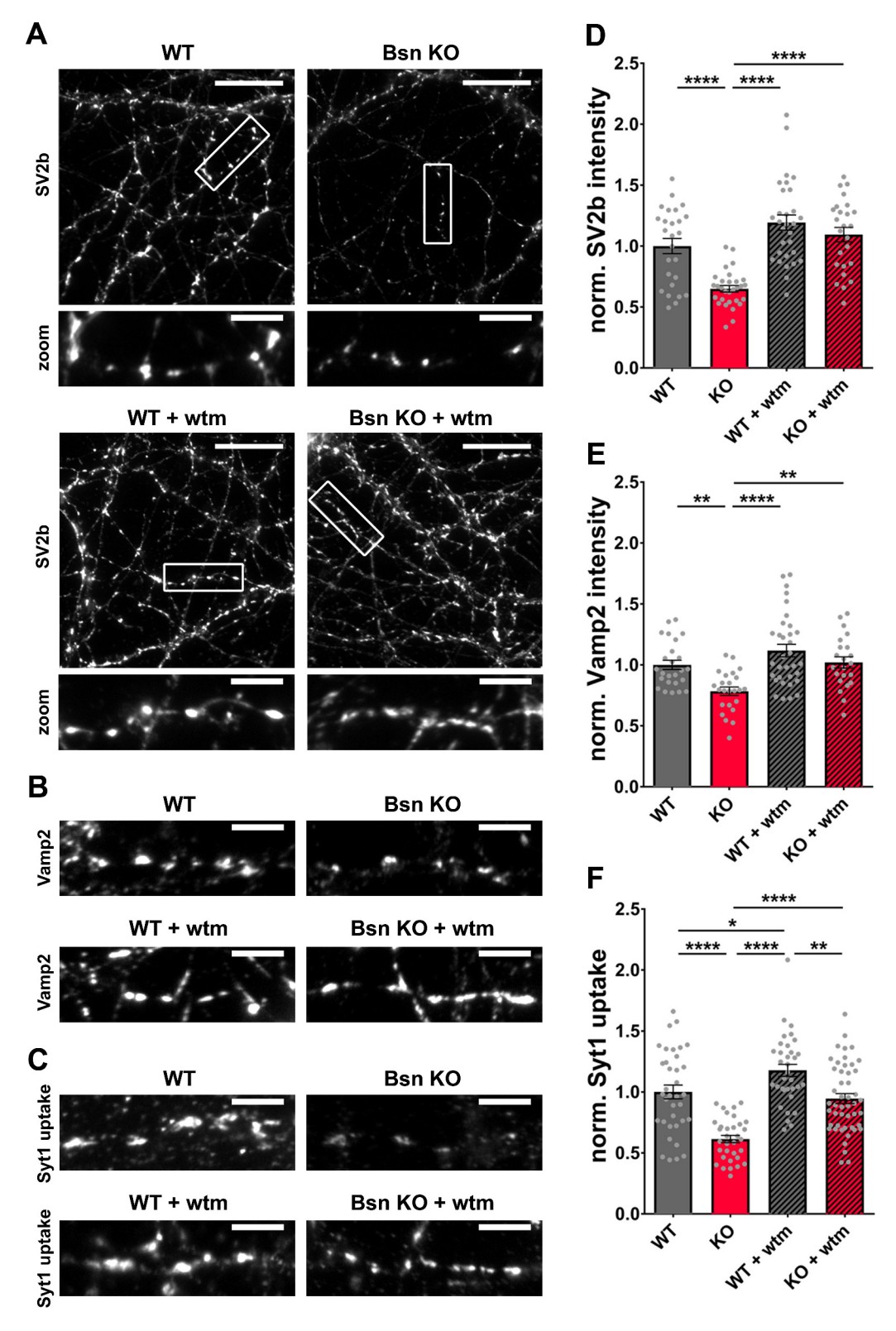

**Figure 4.** *Bassoon* KO neurons display a loss of SV2b and Vamp2 as well as a reduced uptake of Synaptotagmin1 antibody in an autophagy-dependent manner. (A–C) Representative images of hippocampal neurons from WT and *Bsn* KO mice, either left untreated or treated with 1 µM wortmannin (wtm) for 16 hr, stained with antibodies against SV2b (A) and Vamp2 (B) or live labeled with Synaptotagmin1 antibody (Syt1 uptake) (C). (D–F) Quantification of the normalized fluorescence intensities of SV2b (D) (WT = 1.00 ± 0.062, n = 26 images, 3 independent experiments; Bsn KO = 0.65 ± 0.027, n = 30

*Figure 4 continued on next page*

*Figure 4 continued*

images, 3 independent experiments; WT + wtm = 1.19 ± 0.063, n = 30 images, 3 independent experiments; Bsn KO + wtm = 1.09 ± 0.059, n = 25 images, 3 independent experiments; p****<0.0001, p****<0.0001, p****<0.0001), Vamp2 (E) (WT = 1.00 ± 0.038, n = 25 images, 3 independent experiments; Bsn KO = 0.79 ± 0.034, n = 24 images, 3 independent experiments; WT + wtm = 1.12 ± 0.052, n = 33 images, 3 independent experiments; Bsn KO + wtm = 1.02 ± 0.046, n = 22 images, 3 independent experiments; p**=0.0077, p****<0.0001, p**=0.0042) and Syt1 uptake (F) (WT = 1.00 ± 0.056, n = 36 images, 4 independent experiments; Bsn KO = 0.61 ± 0.030, n = 33 images, 4 independent experiments; WT + wtm = 1.18 ± 0.049, n = 34 images, 4 independent experiments; Bsn KO + wtm = 0.95 ± 0.041, n = 48 images, 4 independent experiments; p****<0.0001, p*=0.0429, p****<0.0001, p****<0.0001, p**=0.0017). Scale bars: 20 μm (A), 5 μm (A zoom, B and C). Error bars represent SEM. Data points represent images. ANOVA Tukey's multiple comparisons test was used to evaluate statistical significance.

The online version of this article includes the following figure supplement(s) for figure 4:

**Figure supplement 1.** SAR405 blocks autophagy induction in *Bassoon* KO neurons and rescues SV2b levels.

**Figure supplement 2.** Synaptotagmin1 levels are not decreased in *Bassoon* KO neurons.

## *Bassoon* knockout neurons show an accumulation of autophagic vacuoles in presynaptic terminals and an autophagy-dependent decrease in SV numbers

High numbers of RFP-LC3 puncta in axons from *Bsn* KO neurons suggest that the loss of Bassoon leads to increased autophagosome formation, as previously observed (*Okerlund et al., 2017*). Given the presynaptic localization of Bassoon and the ubiquitination of presynaptic proteins (*Figure 3*), we hypothesized that autophagosomes are being formed locally at the presynaptic terminal to engulf SV-related cargos. To explore this hypothesis, we used cryo-electron microscopy (EM) to quantify the presence of double-membraned autophagic vacuoles (AVs) in WT and *Bsn* KO boutons, as described previously (*Hoffmann et al., 2019*). Consistent with our studies using fluorescent micros- copy (*Okerlund et al., 2017*), significantly more AVs per presynaptic bouton were detected in *Bsn* KO compared to WT neurons (*Figure 5A and C*). This effect was rescued by the addition of wort- mannin (*Figure 5A,B and C*). Intriguingly, AVs filled with SV-sized vesicles were observed in *Bsn* KO samples, indicating a possible wholesale removal of SVs (*Figure 5A*, zoom). In contrast, we could detect only small but not significant differences in the number of multivesicular bodies (MVBs) (*Figure 5A,B and D*), a hallmark of the endo-lysosomal pathway (*Raiborg and Stenmark, 2009*). These data provide compelling evidence that Bassoon-regulated ubiquitination of integral presynap- tic membrane proteins leads to their clearance through autophagy rather than the endo-lysosomal system.

Importantly, our EM data also show that the number of SVs per presynaptic area is reduced in *Bsn* KO neurons (*Figure 5A and E*). This is in line with our findings that SV2b and Vamp2 levels and Syt1 uptake are decreased in neurons lacking Bassoon compared to WT boutons (*Figure 4*). Con- ceptually, these data indicate that Bassoon normally controls the tagging of SV proteins with ubiqui- tin and their clearance through the autophagy system. Consistent with this concept, we observed that treating neurons with wortmannin to inhibit autophagy prior to EM processing increased the number of SVs per terminal area in *Bsn* KO and WT neurons (*Figure 5A,B and E*) further supporting the argument that autophagy plays a role in the maintenance of SVs within presynaptic boutons.

## Bassoon deficiency-dependent autophagy leads to an overall younger pool of SV2

The depletion of Bassoon does not only lead to increased ubiquitination and a loss of SV proteins (*Figure 4*) but simultaneously to increased autophagy levels. Previous studies have shown that the induction of autophagy through the naturally occurring polyamine spermidine is able to expand the lifespan of various animal models (*Eisenberg et al., 2009*). Along these lines, we posited that the increased autophagy in boutons lacking Bassoon would not only increase the rate of SV clearance but also lead to an overall younger pool of SV proteins. Given that SV2 levels are substantially lower in *Bsn* KO boutons, we explored whether this is related to increased protein turnover and accord- ingly a shift in the age of SV2 molecules. This was accomplished by creating a fusion protein between SV2 and a recombinant version of GFP that changes its emission from blue to red with time (*Subach et al., 2009*). For these studies, we chose a medium fluorescent timer (*Subach et al., 2009*) (mFT-SV2) that changes its fluorescence with a maximum of blue intensity after 21 hr and a maximum

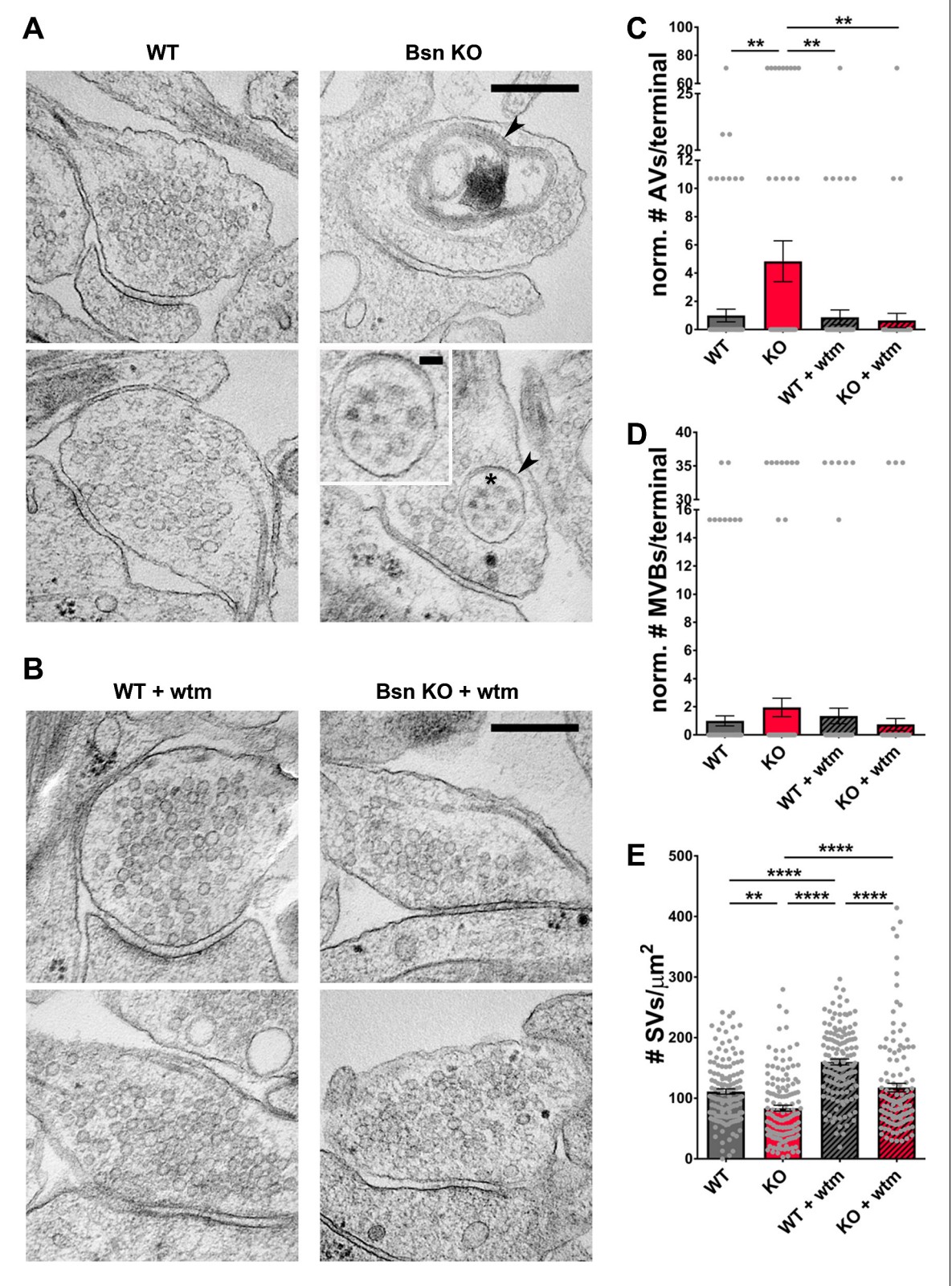

**Figure 5.** Loss of Bassoon leads to the accumulation of autophagic vacuoles and the autophagy-dependent decrease in SV pool size. (A–B) Representative EM micrographs from WT and *Bsn* KO neurons, either left untreated (A) or treated with 1 μM wortmannin (wtm) for 16 hr (B). Arrowheads indicate double-membraned AVs and asterisk indicates ~50 nm vesicles within AV, the latter are also depicted in the zoom. (C–E) Quantification of the normalized number of autophagic vacuoles (AVs) (C) (WT = 1.00 ± 0.453, n = 178 synapses, 2 independent experiments; Bsn

*Figure 5 continued on next page*

*Figure 5 continued*

KO = 4.84 ± 1.448, n = 143 synapses, 2 independent experiments; WT + wtm = 0.87 ± 0.521, n = 143 synapses, 2 independent experiments; Bsn KO + wtm = 0.65 ± 0.507, n = 143 synapses, 2 independent experiments; p\*\*=0.0035, p\*\*=0.0044, p\*\*=0.0022) and multi-vesicular bodies (MVBs) per presynaptic terminal (D) (WT = 1.00 ± 0.355, n = 178 synapses, 2 independent experiments; Bsn KO = 1.95 ± 0.656, n = 143 synapses, 2 independent experiments; WT + wtm = 1.35 ± 0.556, n = 143 synapses, 2 independent experiments; Bsn KO + wtm = 0.75 ± 0.427, n = 143 synapses, 2 independent experiments) and synaptic vesicles (SVs) per terminal area (E) (WT = 111.20 ± 4.416, n = 137 synapses, 2 independent experiments; Bsn KO = 83.63 ± 4.913, n = 132 synapses, 2 independent experiments; WT + wtm = 160.00 ± 4.828, n = 139 synapses, 2 independent experiments; Bsn KO + wtm = 117.70 ± 7.042, n = 124 synapses, 2 independent experiments; p\*\*=0.0014, p\*\*\*\*<0.0001, p\*\*\*\*<0.0001, p\*\*\*\*<0.0001, p\*\*\*\*<0.0001). Scale bars: 300 nm, 50 nm (A zoom). Error bars represent SEM. Data points represent synapses. ANOVA Tukey's multiple comparisons test was used to evaluate statistical significance.

of red intensity after 197 hr. Hippocampal neurons from *Bsn* KO and WT animals, expressing FU-mFT-SV2, show a punctate pattern of both red and blue fluorescence (*Figure 6*). Quantifying the ratio between old (red) and young (blue) SV2, we were able to detect a significant decrease of the red/blue ratio in *Bsn* KO compared to WT neurons (*Figure 6A and B*). This difference is also visible in frequency histograms (*Figure 6C*). These data indicate that indeed the SV2 pool is shifted towards younger proteins in *Bsn*-deficient synapses.

## Parkin is a key regulator of Bassoon deficiency-triggered presynaptic autophagy and SV pool size

The increase in abundance of ubiquitinated synaptic proteins (*Figure 3*) as well as the ability of UbK$_0$ expression to diminish autophagy in *Bsn* KO neurons (*Figure 2*) suggest that functional poly-ubiquitination of SV proteins is required to create the autophagy phenotype in these neurons. An open question is which E3 ubiquitin ligase might be critical for these Bassoon-dependent phenotypes. Our previous studies of neurons lacking both Bassoon and Piccolo revealed that Siah1 was an important E3 ligase controlling SV pool size, but perhaps not elevated autophagy in these cells (*Okerlund et al., 2017*). In revisiting this problem, we initially sought to confirm these results in hippocampal neurons from *Bsn* KO mice. This was accomplished by first verifying in immunoblot analyses that the previously reported shRNAs against *Siah1* (shSiah1) efficiently knocked down *Siah1* (*Okerlund et al., 2017*; *Waites et al., 2013*). In these new experiments, we observed that when ~ 80% of hippocampal neurons were infected with FU-shSiah1, Siah1 levels decreased to ~50% (*Figure 7—figure supplement 1A and C*), indicating that the shRNA is indeed functional. To ensure that any change in autophagy is cell-autonomous and associated with the down regulation of Siah1, *Bsn* KO neurons were either singly or double infected with lentiviruses expressing RFP-LC3, a reporter for autophagy, and FU-shSiah1. Using the soluble eGFP in FU-shSiah1 to identify shSiah1-expressing axons, we observed that reducing Siah1 levels by ~50% was not sufficient to reduce the number of RFP-LC3 puncta present along the axons of neurons lacking Bassoon (*Figure 7A,B and C*). Intriguingly, it slightly reduces the percentage of RFP-LC3 puncta that colocalized with Synaptophysin1 (*Figure 7A,B and E*) indicating that Siah1 does play a role in Bassoon deficiency-triggered autophagy within presynaptic boutons.

Given the ability of RBR-type E3 ubiquitin ligase Parkin to also ubiquitinate SV-associated and other presynaptic proteins (*Chung et al., 2001*; *Huynh et al., 2003*; *Martinez et al., 2017*; *Trempe et al., 2009*), we performed a similar set of experiments with an shRNA generated against *Parkin*. In western blots, our *Parkin* shRNA (shParkin) was able to reduce Parkin protein levels in infected neurons by ~70% (*Figure 7—figure supplement 1B and D*). It also dramatically reduced the number of RFP-LC3 puncta per 10 μm axon (*Figure 8A,B and C*) as well as the colocalization of RFP-LC3 and Synaptophysin1 puncta (*Figure 8A,B and E*, *Figure 1—figure supplement 1E*) to WT levels. These data indicate that Parkin is a potent mediator of presynaptic autophagy caused by the absence of Bassoon. Additionally, lentiviral expression of shParkin significantly increases the number and intensity of Synaptophysin1 puncta along axons compared to the *Bsn* KO alone (*Figure 8A,B and D*).

These data indicate that Bassoon and Parkin work in concert to regulate SV pool size within presynaptic boutons. To explore the relationship of Bassoon and Parkin to SV function and turnover, we examined whether the knockdown of *Parkin* in *Bsn* KO neurons is sufficient to rescue SV protein levels back to WT levels. Using antibodies to monitor the synaptic levels of the SV proteins, SV2b and

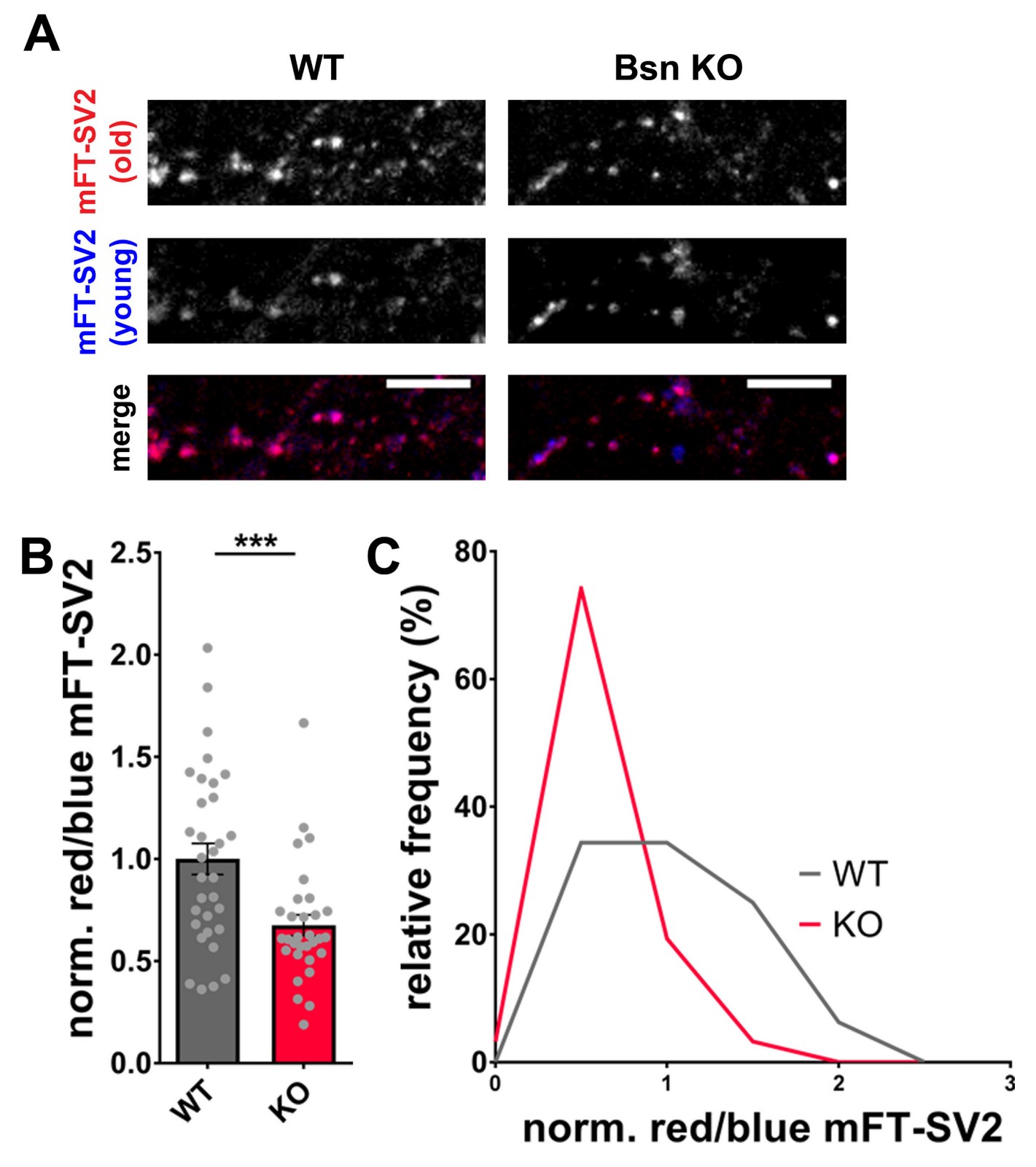

**Figure 6.** Increased autophagy in *Bassoon* KO neurons leads to a younger SV2 protein pool. (**A**) Representative images of hippocampal neurons from WT and *Bsn* KO mice expressing medium fluorescent timer (mFT)-tagged SV2 protein (FU-mFT-SV2). The mFT changes its color from blue to red over time. (**B–C**) Quantification of the normalized red/blue ratio (fluorescence intensities), depicted in a bar graph (**B**) (WT = 1.00 ± 0.076, n = 32 images, 3 independent experiments; Bsn KO = 0.68 ± 0.051, n = 31 images, 3 independent experiments, p***=0.0008) and as a histogram (**C**). Note that *Bsn* KO

*Figure 6 continued on next page*

*Figure 6 continued*

neurons have significantly more blue, thus younger SV2 pools. Scale bars: 10 μm. Error bars represent SEM. Data points represent images. Unpaired t test was used to evaluate statistical significance.

Vamp2, both hyper-ubiquitinated in the absence of Bassoon (*Figure 3*), we could show a significant increase in SV2b and Vamp2 levels in *Bsn* KO neurons expressing shParkin (*Figure 9A,B,D and E*). Furthermore, in Syt1 uptake experiments, the depletion of the E3 ligase Parkin in *Bsn* KO neurons fully prevented the reduced cycling of SV pools (*Figure 9C and F*). These latter data suggest that Parkin plays a fundamental role in the ubiquitin-dependent clearance of SV proteins and thus the integrity and maintenance of functional SV pools.

## Discussion

Although essential for the integrity of the synapse, mechanisms regulating the targeted clearance of SV proteins are not well understood. In this study, we investigate mechanisms regulating the turn-over of SV proteins using the loss of the AZ protein Bassoon as a driver of local autophagy (*Okerlund et al., 2017*). Specifically, our data suggest that Bassoon negatively regulates ubiquitination of a number of SV proteins and that the enhanced ubiquitination of presynaptic proteins is a potent driver of presynaptic autophagy in *Bsn*-deficient neurons, leading to smaller pools of SVs with apparently younger proteins. Our findings also suggest that two E3 ligases, Siah1 and Parkin, become overactive in boutons lacking Bassoon and contribute to the enhanced clearance of SV proteins via the autophagy degradative system, implying that they normally operate to remove non-functional proteins, a feature compromised in neurodegenerative disorders such as Parkinson's disease.

Although previous studies had shown that Bassoon was able to modulate the activity of Atg5, a protein critical for the generation of autophagosomal membranes (*Okerlund et al., 2017*), several lines of evidence collected in this study indicate that the enhanced autophagy seen in boutons lacking Bassoon requires the enhanced ubiquitination of SV-associated proteins. These include data showing that blocking poly-ubiquitination by the over-expression of UbK$_0$ (*Figure 2*; *Okerlund et al., 2017*; *Waites et al., 2013*) or the addition of the E1 inhibitor ziram (*Figure 2—figure supplement 1*; *Chou et al., 2008*; *Rinetti and Schweizer, 2010*) suppresses autophagy in boutons lacking Bassoon (*Figure 2—figure supplement 1*). Moreover, over-expressing the amino-terminal ZnFs of Bassoon, known to bind and inhibit the activity of the E3 ligase Siah1 (*Waites et al., 2013*) not only suppressed autophagy in *Bsn* KO neurons, but restored the number of Synaptophysin1 puncta (*Figure 1*). Furthermore, knocking down either *Siah1* or *Parkin*, two E3 ubiquitin ligases known to ubiquitinate SV-associated and other presynaptic proteins (*Chung et al., 2001*; *Huynh et al., 2003*; *Kabayama et al., 2017*; *Martinez et al., 2017*; *Sassone et al., 2017*; *Trempe et al., 2009*; *Wheeler et al., 2002*), also impaired the induction of autophagy in *Bsn* KO neurons (*Figures 7* and *8*). Intriguingly, while the loss of Bassoon function leads to the selective activation of autophagy and reduction in SV pool size, synapse integrity is not compromised (*Figure 5*). This is in stark contrasts to the situation in neurons, in which the expression of both Bassoon and Piccolo are reduced. Here, not only SV pools are drastically reduced, but also synaptic integrity is disrupted (*Okerlund et al., 2017*; *Waites et al., 2013*). This later situation also involves enhanced poly-ubiquitination of presynaptic proteins and the overactivation of the E3 ligase Siah1 (*Okerlund et al., 2017*; *Waites et al., 2013*). This additional effect seems to be attributable to loss of Piccolo function, which was recently shown to regulate the maintenance of SV pools through Rab5 and Rab7, components of the endo-lysosomal system (*Ackermann et al., 2019*).

Several studies point to the importance of autophagy during health and disease. For instance, reduced autophagy leads to neurodegeneration (*Hara et al., 2006*; *Komatsu et al., 2006*) while enhanced autophagy is correlated with an extended lifespan and an attenuation of age-related memory impairment (*Gupta et al., 2016*; *Vijayan and Verstreken, 2017*). Defining its role and determining potential substrates is essential to understand the impact that autophagy has on the integrity of the synapse.

To better understand how Bassoon regulates SV pool size, we performed mass spectrometry analyses of synaptosomes purified from WT and *Bsn* KO mice. These data revealed a remarkable 2-

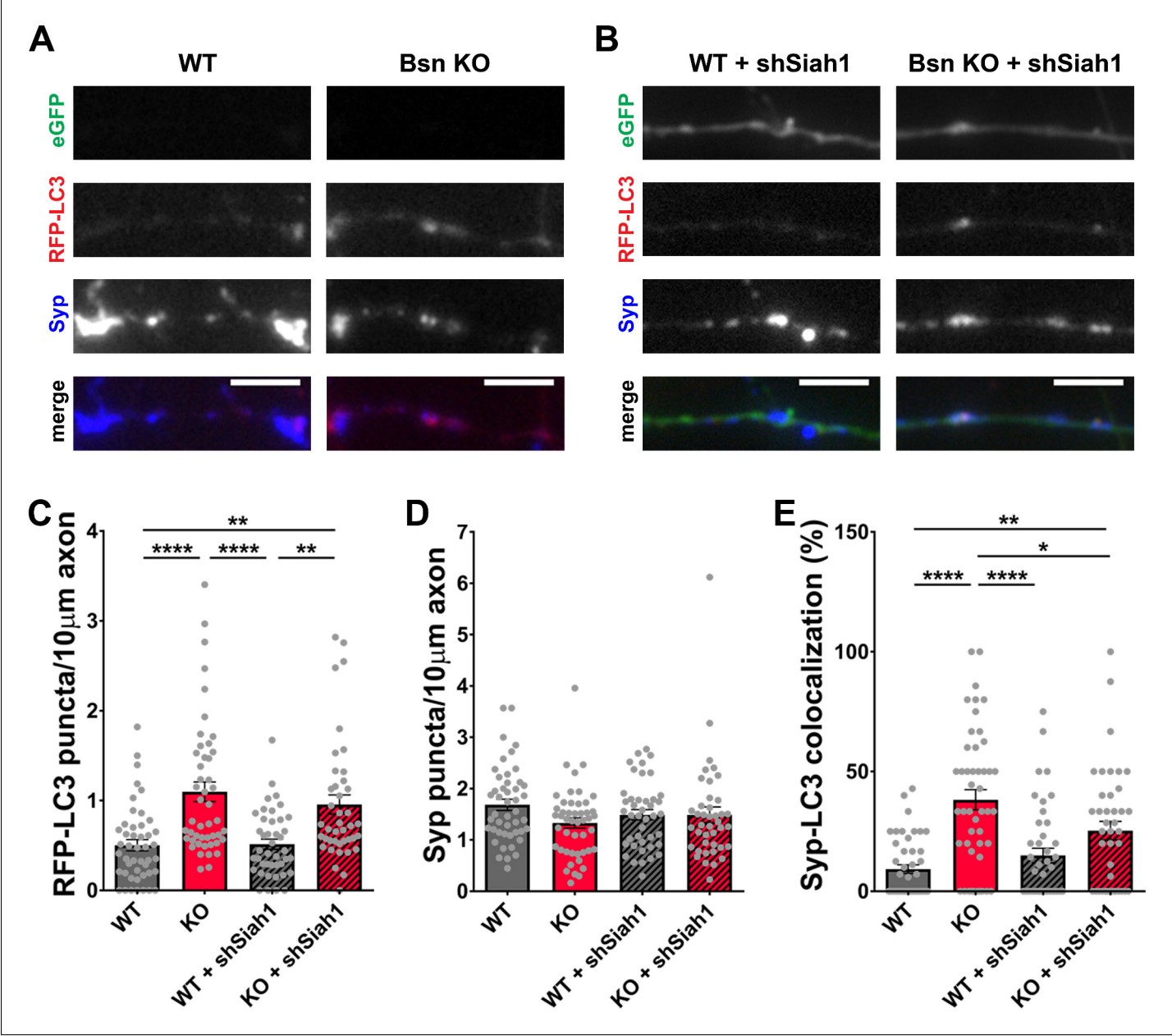

**Figure 7.** Siah1-ubiquitination contributes to increased presynaptic autophagy in *Bassoon* KO neurons. (A–B) Representative images of hippocampal neurons from WT and *Bsn* KO mice expressing FU-RFP-LC3 only (A) or FU-RFP-LC3 and shRNA against *Siah1* (FU-shSiah1) (B) that were fixed and stained with antibodies against Synaptophysin1 (Syp). FU-shSiah1 vector co-expresses a soluble eGFP that enables the identification and quantification of axons from infected neurons only. (C–E) Quantification of the number of RFP-LC3 puncta per 10 μm axon (C) (WT = 0.50 ± 0.063, n = 46 axons, 3 independent experiments; Bsn KO = 1.10 ± 0.108, n = 47 axons, 3 independent experiments; WT + shSiah1 = 0.52 ± 0.056, n = 42 axons, 3 independent experiments; Bsn KO + shSiah1 = 0.96 ± 0.106, n = 41 axons, 3 independent experiments; p****<0.0001, p**=0.0019, p****<0.0001, p**=0.0035), the number of Syp1 puncta per 10 μm axon (D) (WT = 1.68 ± 0.109, n = 46 axons, 3 independent experiments; Bsn KO = 1.33 ± 0.100, n = 47 axons, 3 independent experiments; WT + shSiah1 = 1.49 ± 0.101, n = 42 axons, 3 independent experiments; Bsn KO + shSiah1 = 1.49 ± 0.152, n = 41 axons, 3 independent experiments) and the colocalization of RFP-LC3 and Syp1 (E) (WT = 9.23 ± 1.809, n = 46 axons, 3 independent experiments; Bsn KO = 38.19 ± 4.211, n = 47 axons, 3 independent experiments; WT + shSiah1 = 14.93 ± 3.046, n = 42 axons, 3 independent experiments; Bsn KO + shSiah1 = 25.28 ± 3.902, n = 41 axons, 3 independent experiments; p****<0.0001, p**=0.0056, p****<0.0001, p*=0.0379). Scale bars: 5 μm. Error bars represent SEM. Data points represent axons. ANOVA Tukey's multiple comparisons test was used to evaluate statistical significance.
The online version of this article includes the following figure supplement(s) for figure 7:

**Figure supplement 1.** Knockdown efficiency of shSiah1 and shParkin.

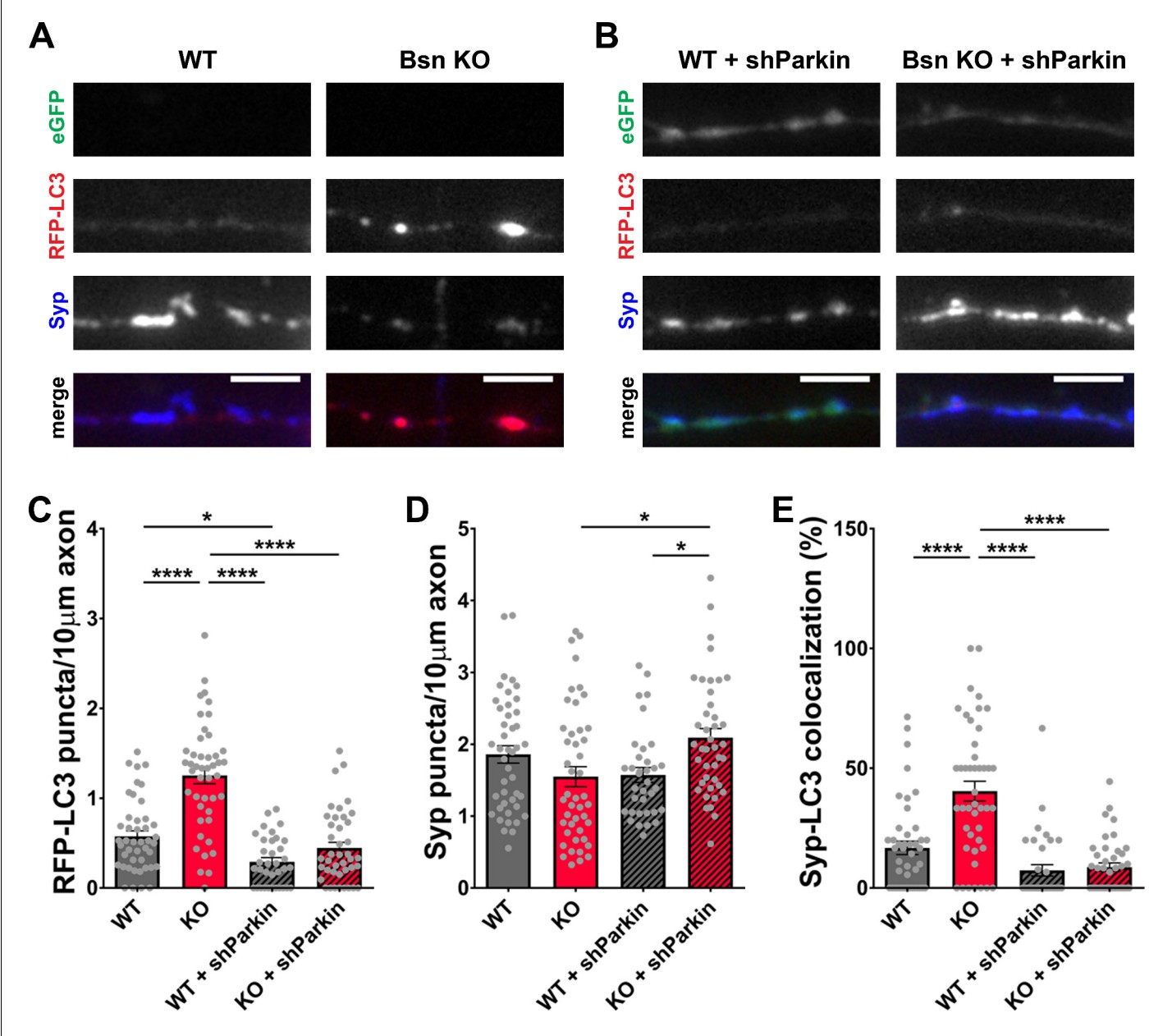

**Figure 8.** Knockdown of Parkin rescues the autophagy phenotype in *Bassoon* KO neurons. (A–B) Representative images of hippocampal neurons from WT and *Bsn* KO mice expressing FU-RFP-LC3 only (A) or FU-RFP-LC3 and shRNA against *Parkin* (FU-shParkin) (B) that were fixed and stained with antibodies against Synaptophysin1 (Syp). FU-shParkin vector co-expresses a soluble eGFP that enables the identification and quantification of axons from infected neurons only. (C–E) Quantification of the number of RFP-LC3 puncta per 10 µm axon (C) (WT = 0.58 ± 0.060, n = 44 axons, 3 independent experiments; Bsn KO = 1.25 ± 0.090, n = 45 axons, 3 independent experiments; WT + shParkin = 0.29 ± 0.046, n = 35 axons, 3 independent experiments; Bsn KO + shParkin = 0.45 ± 0.063, n = 41 axons, 3 independent experiments; p****<0.0001, p*=0.0285, p****<0.0001, p****<0.0001), the number of Syp1 puncta per 10 µm axon (D) (WT = 1.86 ± 0.121, n = 44 axons, 3 independent experiments; Bsn KO = 1.55 ± 0.139, n = 45 axons, 3 independent experiments; WT + shParkin = 1.57 ± 0.104, n = 35 axons, 3 independent experiments; Bsn KO + shParkin = 2.09 ± 0.127, n = 41 axons, 3 independent experiments; p*=0.0119, p*=0.0302) and the colocalization of RFP-LC3 and Syp1 (E) (WT = 16.75 ± 2.82, n = 44 axons, 3 independent experiments; Bsn KO = 40.42 ± 4.134, n = 45 axons, 3 independent experiments; WT + shParkin = 7.38 ± 2.382, n = 35 axons, 3 independent experiments; Bsn KO + shParkin = 8.72 ± 1.736, n = 41 axons, 3 independent experiments; p****<0.0001, p****<0.0001, p****<0.0001). Scale bars: 5 µm. Error bars represent SEM. Data points represent axons. ANOVA Tukey's multiple comparisons test was used to evaluate statistical significance.

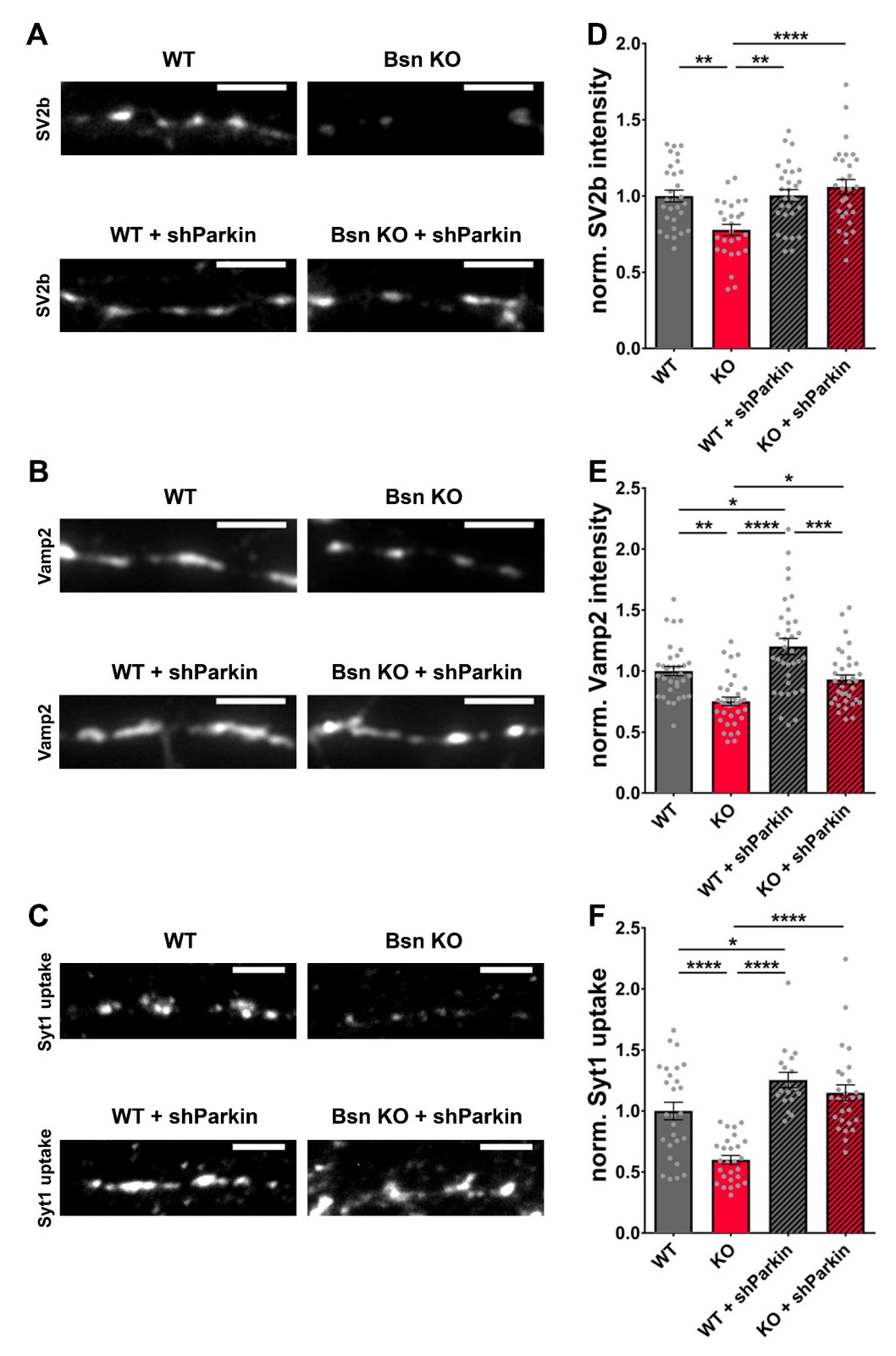

**Figure 9.** Decreased SV2b, Vamp2 and Synaptotagmin1 antibody uptake in *Bassoon* KO neurons are rescued by the downregulation of Parkin. (A–C) Representative images of hippocampal neurons from WT and *Bsn* KO mice, either uninfected or expressing shRNA against Parkin (FU-shParkin) that were fixed and stained with antibodies against SV2b (A) and Vamp2 (B) or live labeled with Synaptotagmin1 antibody (Syt1 uptake) (C). (D–F) Quantification of the normalized fluorescence intensities of SV2b (D) (WT = 1.00 ± 0.039, n = 29 images, 3 independent experiments; Bsn

*Figure 9 continued on next page*

*Figure 9 continued*

KO = 0.78 ± 0.037, n = 27 images, 3 independent experiments; WT + shParkin = 1.00 ± 0.039, n = 29 images, 3 independent experiments; Bsn KO + shParkin = 1.06 ± 0.049, n = 29 images, 3 independent experiments; p**=0.0014, p**=0.0012, p****<0.0001), Vamp2 (**E**) (WT = 1.00 ± 0.038, n = 34 images, 3 independent experiments; Bsn KO = 0.75 ± 0.035, n = 35 images, 3 independent experiments; WT + shParkin = 1.20 ± 0.066, n = 34 images, 3 independent experiments; Bsn KO + shParkin = 0.93 ± 0.038, n = 35 images, 3 independent experiments; p**=0.0011, p*=0.0126, p****<0.0001, p*=0.0319, p***=0.0003) and Syt1 uptake (**F**) (WT = 1.00 ± 0.072, n = 27 images, 3 independent experiments; Bsn KO = 0.60 ± 0.035, n = 27 images, 3 independent experiments; WT + shParkin = 1.25 ± 0.062, n = 18 images, 3 independent experiments; Bsn KO + shParkin = 1.15 ± 0.065, n = 27 images, 3 independent experiments; p****<0.0001, p*=0.0343, p****<0.0001, p****<0.0001). Scale bars: 5 µm. Error bars represent SEM. Data points represent images. ANOVA Tukey's multiple comparisons test was used to evaluate statistical significance.

to 4-fold increase in ubiquitination levels of several SV proteins when Bassoon is depleted (*Figure 3*). Levels of ubiquitinated postsynaptic proteins such as SynGAP were not changed in these mice, supporting earlier conclusions that Bassoon is regulating autophagy and/or ubiquitination on a local presynaptic level (*Okerlund et al., 2017*). One limitation of our mass spectrometry findings is that we cannot distinguish between mono-ubiquitination and poly-ubiquitination, nor can we identify which lysine residues in the ubiquitin poly-peptide are used to create longer ubiquitin chains. Given the high variety of how poly-ubiquitin molecules can be attached to any given substrate (*Kwon and Ciechanover, 2017*), we cannot discriminate whether mainly changes in K63- or K48-poly-ubiquitination do occur. For instance, while Synaptagmin1 is highly ubiquitinated in *Bsn* KO neurons, this modification did not lead to a significant loss of protein (*Figure 4—figure supplement 2B*), making it particularly important to study more synaptic proteins and their unique ubiquitination patterns in future studies.

Nonetheless, the requirement for ubiquitination for the induction of autophagy and the loss of SVs in *Bsn* KO boutons in combination with the appearance of ubiquitinated SV proteins imply that at least some of the attached poly-ubiquitin chains must be K63-conjugated. Given the cytoplasmic location of the ubiquitin machinery (*Hoffmann et al., 2019*), one can anticipate that loss of Bassoon function increases the poly-ubiquitination on cytoplasmic sites of various SV proteins. Consistently, we found that the ubiquitinated residues on each analyzed peptide occurred on the cytoplasmic site of the corresponding protein (*Figure 3*). For example, the ubiquitination sites on SV2a and SV2b (SV2a$_{143}$ and SV2b$_{333}$) were located on their cytoplasmic tails (*Bartholome et al., 2017*), further strengthening our hypothesis that the observed ubiquitination of SV proteins tags them for degradation. Additionally, we could detect ubiquitinated peptides from E2N, a ubiquitin conjugating enzyme, known to be involved in K63-poly-ubiquitination (*David et al., 2010*). This suggests that the absence of Bassoon selectively activates E2N, which then transfers ubiquitin to substrates in the context of specific E3 ligases, a concept that supports our hypothesis that the autophagy phenotype in *Bsn* KO neurons is attributable to increased K63-ubiquitination.

Recent evidence has shown that ROS-damaged SV proteins can enter the autophagy pathway (*Hoffmann et al., 2019*). Thus, a fundamental question raised by our study is whether increasing autophagy through Bassoon loss of function also leads to the clearance of SV proteins. Light level quantification of SV2b and Vamp2 revealed a significant decrease of both proteins (*Figure 4*) in *Bsn* KO neurons in an autophagy-dependent manner. Similarly, the appearance of autophagic vacuoles containing ~50 nm vesicles as assessed by cryo-EM and loss of SVs in *Bsn* KO boutons (*Figure 5*) makes a strong argument that SV proteins or even whole SVs can enter the autophagic pathway for their clearance. The latter observation is not surprising, as the knockdown of Bassoon in a Piccolo KO mouse model drives the loss of SVs in boutons of cortical cultures as well (*Mukherjee et al., 2010*). These findings are consistent with previous data showing a severe reduction of eGFP-SV2 intensities in the boutons lacking Bassoon and Piccolo (*Okerlund et al., 2017*; *Waites et al., 2013*) and further indicate that (ubiquitinated) SV proteins could become cargo of Bassoon deficiency-triggered autophagy.

One prediction of enhanced autophagy in neurons lacking Bassoon is that primarily old proteins are being removed. Consistently, we found that, when SV2 was tagged with a medium fluorescent timer (*Subach et al., 2009*), a significantly younger pool of SV2 could be detected at synapses from *Bsn* KO neurons compared to WT neurons (*Figure 6*). Accordingly, the absence of Bassoon seems to enhance autophagic degradation of SV proteins.

Another important question raised by our study is which degradative system is preferentially turned on in our model system? Given that inhibition of the proteasome induces autophagy to compensate for the reduced clearance of soluble proteins (*Ding et al., 2007*) and that autophagosomes can fuse with endosomes (*Klionsky et al., 2014*), it becomes clear that these three degradative pathways are strongly co-regulated to ensure synaptic health (*Lamark and Johansen, 2010*; *Lilienbaum, 2013*; *Vijayan and Verstreken, 2017*). This interdependence makes it difficult to distinguish the contribution of any one of them. To meet this challenge, we quantified the presence of both AVs as a measure of autophagy (*Ericsson, 1969*; *Klionsky et al., 2012*) and MVBs as a measure of the endo-lysosomal machinery (*Raiborg and Stenmark, 2009*) per presynaptic terminal in WT and *Bsn* KO neurons. Here, we could detect an increase of AVs in *Bsn* KO neurons, with no profound changes in the number of MVBs (*Figure 5*). This suggests that Bassoon deficiency regulates the clearance of synaptic proteins primarily through the autophagy rather than the endo-lysosomal system. At present, we cannot rule out potential roles for Bassoon in other degradative pathways such as the proteasomal or the endo-lysosomal system; an interesting topic for future studies.

In a previous study, we found that the loss of SVs and synaptic integrity associated with the loss of Piccolo and Bassoon required both poly-ubiquitination and the RING-type E3 ligase Siah1 (*Waites et al., 2013*). A follow-up study revealed that the loss of both active zone proteins also triggered the induction of autophagy, but found no strong evidence that this required Siah1 (*Okerlund et al., 2017*), perhaps because this condition activated multiple E3 ligases generating ubiquitinated substrates that promote autophagy independent of Siah1. The subsequent finding that Bassoon, rather than Piccolo, was a local active zone regulator of autophagy, allowed us in the present study to revisit the question of whether substrates generated by Siah1 promote presynaptic autophagy in the absence of Bassoon. Here, we found a significant reduction in presynaptic autophagosomes, identified with RFP-LC3 colocalizing with Synaptophysin1, in neurons with a 50% reduction in the expression of Siah1. But, this manipulation did not significantly affect the total pool of autophagosomes that appeared in axons (*Figure 7*), suggesting either that the 50% reduction of Siah1 is not sufficient to block its activity or that other ubiquitin ligases may contribute to this phenotype. The data also indicate that Siah1-mediated ubiquitination of SV proteins, such as Synaptophysin (*Wheeler et al., 2002*), may contribute to the elimination of SV proteins through other degradative systems.

Importantly, our studies show that the RBR-type E3 ligase Parkin is required for elevated autophagy levels in *Bsn* KO mice (*Figure 8*). Parkin not only plays an important role in mitophagy (*Ashrafi et al., 2014*; *Pickrell and Youle, 2015*) but it can also generate K63-poly-ubiquitinated substrates in general (*Olzmann et al., 2007*). Another study showed that, in *Drosophila* neurons, Parkin strongly mediates K6- and K63-poly-ubiquitination (*Martinez et al., 2017*). Increased K6-ubiquitination through Parkin could be explained by the fact that Parkin auto-ubiquitinates itself via K6-ubiquitin for stability regulation (*Durcan et al., 2014*). Besides being involved in mitophagy, there have been reports that Parkin ubiquitinates several substrates that are abundant at the synapse, such as Synphilin-1 (*Chung et al., 2001*), Synaptotagmin11 (*Huynh et al., 2003*), Synaptotagmin4 (*Kabayama et al., 2017*), CDCrel-1 (*Zhang et al., 2000*), Syntaxin5 (*Martinez et al., 2017*) and Endophilin-A (*Murdoch et al., 2016*; *Trempe et al., 2009*). Intriguingly, our mass spectrometry analysis revealed a strong increase in the ubiquitinated Parkin substrate Synaptotagmin11 (*Figure 3*). We further identified increased ubiquitination in other SV proteins (SV2, Vamp2, Synaptotagmin1) as well as components of the SNARE SV fusion complex (SNAP25 and Syntaxin1b), raising the possibility that they are also substrates for Parkin-mediated ubiquitination. Future studies will help resolve these possibilities. Recently, it has also been shown that Parkin can be activated independently of the mitophagy-related PINK1 protein (*Balasubramaniam et al., 2019*). The significant rescue of synaptic autophagy with shParkin and the observation that Parkin is involved in ubiquitination of several synaptic proteins strongly suggest that Parkin is involved in autophagy induced by the absence of Bassoon.

In an attempt to further investigate the role of Parkin in synaptic vesicle autophagy, we were able to rescue SV2b and Vamp2 levels, as well as SV cycling, by knocking down *Parkin* (*Figure 9*). Given that total Synaptotagmin1 levels are not changed, the reduction in the Synaptotagmin1 antibody uptake driven by endogenous network activity observed in *Bsn* KO neurons can be either caused by the decrease of total SV number (*Figure 5*), by the contribution of Bassoon to SV release (*Altrock et al., 2003*) or reloading into release sites (*Hallermann et al., 2010*), or its role in the

recruitment of voltage-dependent $Ca_V2.1$ channels to the presynaptic membrane (*Davydova et al., 2014*). Alternatively, exocytosis could be impaired by the removal of the highly ubiquitinated SNARE proteins, SNAP25 and Syntaxin1b (*Figure 3*). Together these data provide strong evidence that Parkin is one of the major E3 ligases that are required for Bassoon deficiency-triggered SV autophagy. An open question is how Bassoon influences Parkin activity. As for Siah1 (*Waites et al., 2013*), we have performed various biochemical studies to assess whether Bassoon directly interacts with Parkin. However, such an interaction was not detectable in immunoprecipitation experiments performed with either full-length recombinant Bassoon or various of its subdomains (data not shown). Nonetheless, both proteins could be part of a larger assembly, in which Bassoon regulates Parkin activity. Alternatively, the absence of Bassoon could trigger indirectly pathways that enhance Parkin activity. Future experiments will have to address these issues.

In conclusion, our data suggest that SV proteins are indeed cargos of Bassoon deficiency-triggered autophagy and that the E3 ligases Siah1 and more potently Parkin are crucial ligases promoting the ubiquitination of SV proteins and their removal via the autophagy system. Clearly, the overactivation of this system by the loss of Bassoon can have adverse effects on the function of presynaptic boutons, arguing that a balanced regulation is necessary to maintain the functional integrity of SV and other presynaptic proteins. Future studies investigating Parkin's role in SV degradation would be worthwhile given its strong involvement in neurodegenerative diseases such as Parkinson's disease (*Nixon, 2013*; *Pickrell and Youle, 2015*).

Importantly, the data presented here raise several fundamental questions regarding the contribution of other degradative systems, such as the proteasome and the endo-lysosomal system, to the maintenance of presynaptic proteostasis. Previous studies have shown that the proteasome can utilize specific E3 ligase such as SCRAPPER or Fbxo45 to mediate the removal of specific AZ proteins such as RIM1 and Munc13, sculpting the efficiency of synaptic transmission (*Jiang et al., 2010*; *Speese et al., 2003*; *Tada et al., 2010*; *Yao et al., 2007*; *Yi and Ehlers, 2005*). Evidence has also emerged that homeostatic activity-dependent changes in synaptic function engage the endo-lysosomal system for the removal of subsets of SV proteins (*Jin et al., 2018*; *Sheehan et al., 2016*). It has also been found that nutrient starvation can trigger the degradation of a broad range of neuronal as well as synaptic proteins to reduce metabolic demand for cells during periods of stress (*Catanese et al., 2018*). This suggests that these degradative systems function in synchrony to modulate the composition of synaptic proteins to a broad range of physiological conditions. Less clear is whether there is functional redundancy within these systems that can compensate for the loss of one, during for example neurodegenerative disorders. Understanding the fundamental mechanisms that regulate each will be critical to sorting this out. Based on our observations concerning Parkin, we can anticipate that though there are approximately 600 different E3 ligases, each can likely operate in unique ways and on different substrates depending on the microenvironment.

## Materials and methods

### Key resources table

| Reagent type (species) or resource | Designation | Source or reference | Identifiers | Additional information |
|---|---|---|---|---|
| Strain, strain background *Mus musculus* both sexes) | Bassoon knockout mice | Leibniz Institute for Neurobiology, laboratory of Eckart Gundelfinger; *Hallermann et al., 2010*; *Davydova et al., 2014* | Omnibank ES cell line OST486029 by Lexicon Pharmaceuticals (The Woodlands, TX; USA) | Bsn/Bassoon KO mice created by gene trap |
| Strain, strain background *Mus musculus* both sexes) | WT mice (C57BL/6J) | Research Institutes for experimental Medicine (FEM), Charité Berlin | RRID:IMSR_JAX:000664 | |

*Continued on next page*

*Continued*

| Reagent type (species) or resource | Designation | Source or reference | Identifiers | Additional information |
|---|---|---|---|---|
| Transfected construct (Rattus norvegicus) | FU-RFP-LC3 | *Waites et al., 2013*; *Okerlund et al., 2017* | | Lentiviral construct to infect and express autophagy reporter |
| Transfected construct (Rattus norvegicus) | FU-UbK$_0$ | *Waites et al., 2013*; *Okerlund et al., 2017* | | Lentiviral construct to infect and express recombinant ubiquitin |
| Transfected construct Mus musculus | FU-shSiah1 | *Waites et al., 2013*; *Okerlund et al., 2017* | | Lentiviral construct to infect and express shRNA against Siah1 |
| Transfected construct Mus musculus | FU-shParkin | This paper; Viral Core Facility (VCF), Charité Berlin | | Lentiviral construct to infect and express shRNA against Parkin 'gcaaCgtgccAattga aaattcaagagattttcaat TggcacGttgc' |
| Transfected construct Mus musculus | FU-mFT-SV2 | This paper; Viral Core Facility (VCF), Charité Berlin | | medium fluorescent timer from *Subach et al., 2009* |
| tTransfected construct Mus musculus | FU-Bsn609-eGFP | This paper; Viral Core Facility (VCF), Charité Berlin | | Lentiviral construct to infect and express Bsn609 |
| Antibody | anti-Synaptophysin (mouse monoclonal) | Synaptic Systems | Cat# 101011; RRID:AB_887824 | 1:1000 |
| Antibody | anti-Vamp2 (rabbit polyclonal) | Synaptic Systems | Cat# 104202; RRID:AB_887810 | 1:1000 |
| Antibody | anti-SV2b (rabbit polyclonal) | Synaptic Systems | Cat# 119102; RRID:AB_887803 | 1:1000 |
| Antibody | anti-Synaptotagmin1 (mouse monoclonal) | Synaptic Systems | Cat# 105011; RRID:AB_887832 | 1:1000 |
| Antibody | anti-Synaptotagmin1 lumenal (rabbit polyclonal) | Synaptic Systems | Cat# 105103C3; RRID:AB_887829 | 1:70 (Cy3-labeled for live labeling) |
| Antibody | anti-Parkin (mouse monoclonal) | abcam | Cat# ab77924; RRID:AB_1566559 | 1:500 |
| Antibody | anti-Siah1 (rabbit polyclonal) | abcam | Cat# ab203198; RRID:AB_2833241 | 1:500 |
| Antibody | anti-LC3 (rabbit monoclonal) | Cell Signaling Technology | Cat# 12741; RRID:AB_2617131*Updated | 1:1000 |
| Antibody | anti-GAPDH (rabbit monoclonal) | Cell Signaling Technology | Cat# 2118; RRID:AB_561053*Updated | 1:1000 |
| Antibody | anti-Actin (rabbit polyclonal) | Sigma-Aldrich | Cat# A2066; RRID:AB_476693 | 1:1000 |
| Antibody | Alexa Fluor 647 goat anti-mouse secondary antibody | ThermoFisher Scientific | Cat# A21236; RRID:AB_2535805 | 1:1000 |
| Antibody | Alexa Fluor 647 goat anti-rabbit secondary antibody | ThermoFisher Scientific | Cat# A21245; RRID:AB_2535813 | 1:1000 |
| Antibody | Alexa Fluor 568 goat anti-rabbit secondary antibody | ThermoFisher Scientific | Cat# A11036; RRID:AB_10563566 | 1:2000 |

*Continued on next page*

*Continued*

| Reagent type (species) or resource | Designation | Source or reference | Identifiers | Additional information |
|---|---|---|---|---|
| Antibody | Cy3 donkey anti-rabbit secondary antibody | Dianova/Jackson ImmunoResearch | Cat# 711-165-152; RRID:AB_2307443 | 1:2000 |
| Chemical compound, drug | Wortmannin | InvivoGen | Cat# tlrl-wtm; CAS Number 19545-26-7 | 1 µM, 16 hr |
| Chemical compound, drug | Ziram PESTANAL | Sigma-Aldrich | Cat# 45708; CAS Number 137-30-4 | 1 µM, 16 hr |
| Chemical compound, drug | SAR405 | Hycultec | Cat# HY12481A; CAS Number 1946010–7902 | 1 µM, 16 hr |
| Commercial assay or kit | Ubiquitin Remnant Motif (K-ε-GG) Kit | Cell Signaling Technology | Cat# 5562 | |
| Commercial assay or kit | Syn-PER Synaptic Protein Extraction Reagent | Thermo Scientific | Cat# 87793 | |
| Software, algorithm | OpenView | *Tsuriel et al., 2006* | N/A | Written by Prof. Noam Ziv |
| Software, algorithm | ImageJ | National Institute of Health | https://Imagej.nih.gov/ RRID:SCR_003070 | |
| Software, algorithm | Prism | GraphPad Software | https://www.graphpad.com/ RRID:SCR_002798 | |
| Software, algorithm | Mascot software version 2.6.1 | Matrix Science Ltd., London, UK | RRID:SCR_014322 | |
| Software, algorithm | MaxQuant software version 1.6.0.1 | Max-Planck-Institute of Biochemistry | RRID:SCR_014485 | |

## Animals

*Bassoon (Bsn)* knockout (KO) mice, created by gene trap resulting in a knockout of *Bsn* in most neurons with some residual protein expressed, e.g. in the cochlea (*Frank et al., 2010*; *Hallermann et al., 2010*; *Jing et al., 2013*), are called *Bsn* KO mice throughout this study. These animals were used to obtain tissues and cells used in this study. Breeding of animals and experiments using animal material were carried out in accordance with the European Communities Council Directive (2010/63/EU) and approved by the local animal care committees of Sachsen-Anhalt or the animal welfare committee of Charité Medical University and the Berlin state government (protocol number: T0036/14, O0208/16).

## Lentiviral vectors

The lentiviral vectors FU-RFP-LC3, FU-UbK$_0$ and FU-shSiah1 were published previously (*Okerlund et al., 2017*; *Waites et al., 2013*). In brief, FU-RFP-LC3 expresses RFP-tagged LC3 under the ubiquitin promotor, FU-UbK$_0$ expresses soluble eGFP and a recombinant ubiquitin without lysine residues under the ubiquitin promotor and FU-shSiah1 expresses soluble eGFP under the ubiquitin promotor as well as an shRNA against *Siah1* under a U6 promotor. The vector FU-shParkin co-expresses soluble eGFP under a ubiquitin promotor as well as an shRNA against *Parkin* under a U6 promotor ('gcaaCgtgccAattgaaaattcaagagattttcaatTggcacGttgc') (*Cortese et al., 2016*), adapted for mouse. The vector FU-mFT-SV2 was designed to express a medium fluorescent timer (*Subach et al., 2009*) linked to the cytoplasmic tail of SV2 via a Glycine linker. The vector FU-Bsn609-eGFP expresses the N-terminal 609 amino acids of Bassoon tagged at the C-terminus with eGFP (*Dresbach et al., 2003*) under the ubiquitin promotor; the fusion peptide is efficiently targeted

to presynaptic boutons (*Altrock et al., 2003*; *Figure 1*). All vectors are based on the commercially available vector FUGW (Addgene).

## Preparation of cultured hippocampal neurons and viral infections

To assess the efficiency of shRNA-mediated knockdown in western blot experiments, wild type (WT) hippocampi were dissected from mice P0-2 (postnatal day 0–2) brains in cold Hanks' Salt Solution (Millipore, Darmstadt, Germany), followed by a 30 min incubation in enzyme solution (DMEM (Gibco, Thermo Fisher Scientific, Waltham, USA), 3.3 mM Cystein, 2 mM CaCl$_2$, 1 mM EDTA, 20 U/ml Papain (Worthington, Lakewood, USA)) at 37°C. Papain reaction was inhibited by the incubation of hippo-campi in inhibitor solution DMEM, 10% fetal calf serum (FCS) (Thermo Fisher Scientific), 38 mM BSA (Sigma-Aldrich, St. Louis, USA) and 95 mM Trypsin Inhibitor (Sigma-Aldrich) for 5 min. Afterwards, cells were triturated in NBA (Neurobasal-A Medium, 2% B27, 1% Glutamax, 0.2% P/S) (Thermo Fisher Scientific) by gentle pipetting up and down. Dissociated hippocampal neurons were plated directly onto 6-well culture dishes at a density of 25 k per 1 cm$^2$ and maintained in NBA at 37°C, 5% CO$_2$, for 13–15 days in vitro (DIV) before starting experiments. For viral infections the cultures were transduced with lentiviral particles with nearly 100% efficiency of transduction on 1–2 DIV and cells harvested for western blotting at 13–15 DIV.

For live cell imaging and immunocytochemistry, hippocampal neuron cultures were prepared as described previously (*Davydova et al., 2014*; *Okerlund et al., 2017*) from P0-1 *Bsn* KO mice and their wildtype littermates and plated on glass coverslips using the Banker protocol (*Banker and Goslin, 1988*; *Meberg and Miller, 2003*). Briefly, astrocytes from mouse WT cortices P0-1 were seeded on 5 cm plates 5–7 d before neuron preparation to create a 60–70% confluent monolayer of astrocytes on the day of neuronal plating. After trypsin treatment and mechanical trituration, hippo-campal neurons were plated in densities of 35,000 cells per 18 mm diameter coverslip. One hour after plating, coverslips were transferred upside down into dishes containing astrocytes and Neuro-basal A medium supplemented with B27, 1 mM sodium pyruvate, 4 mM Glutamax and antibiotics (P/S). At DIV 1 and DIV 3, Cytosine β-D-arabinofuranoside (Ara-C, Sigma-Aldrich) was added to the cells to reach a final concentration of 1.2 µM. The co-culture was kept in NBA at 37°C, 5% CO$_2$, for 14–15 DIV before starting experiments. For viral infections the cultures were transduced with lentivi-ral particles with nearly 100% efficiency of transduction on 1–2 DIV and experiments were performed at 14–15 DIV.

## Lentivirus production

All lentiviral particles were provided by the Viral Core Facility of the Charité - Universitätsmedizin Berlin (vcf.charite.de) and were prepared as described previously (*Hoffmann et al., 2019*). Briefly, HEK293T cells were cotransfected with 10 µg of shuttle vector, 5 µg of helper plasmid pCMVdR8.9, and 5 µg of pVSV.G with X-tremeGENE 9 DNA transfection reagent (Roche Diagnostics, Mannheim, Germany). Virus containing cell culture supernatant was collected after 72 hr and filtered for purifica-tion. Aliquots were flash-frozen in liquid nitrogen and stored at −80°C.

## Immunocytochemistry of hippocampal neurons

Primary hippocampal neurons from *Bsn* KO and WT mice were fixed at 14–15 DIV with 4% parafor-maldehyde (PFA) in phosphate buffered saline (PBS) for 4 min and washed twice with PBS (10 min each). Afterwards, cells were quenched with 25 mM glycine in PBS for 20 min and permeabilized with blocking solution (2% bovine serum albumin (BSA), 5% normal goat serum (NGS) in PBS) + 0.2% Triton X100 for 1 hr. Then, neurons were incubated with primary antibodies, diluted in blocking solution, for 1 hr at RT. The following antibodies were used: primary antibodies against Synaptophy-sin1 (1:1000; mouse; Synaptic Systems, Göttingen, Germany; Cat# 101011), Vamp2 (1:1000; rabbit; Synaptic Systems; Cat# 104202), SV2b (1:1000; rabbit; Synaptic Systems; Cat# 119102) and Synapto-tagmin1 (1:1000; mouse; Synaptic Systems; Cat# 105011). Afterwards cells were washed three times in blocking solution for 10 min each, incubated with the secondary antibody (Alexa Fluor 647 goat-anti mouse IgG, Cat# A-21236, Alexa Fluor 647 goat-anti rabbit IgG, Cat# A-21245, Alexa Fluor 568 goat-anti rabbit IgG, Cat# A-11036, Thermo Fisher Scientific; Cy3 donkey-anti rabbit IgG, Cat# 711-165-152, Dianova, Hamburg, Germany) diluted in blocking solution 1:1000 or 1:2000 for 60 min and washed once with blocking solution and twice with PBS 10 min each. Finally, coverslips were dipped

in H$_2$O and mounted in ProLong Diamond Antifade Mountant (Thermo Fisher Scientific). To inhibit autophagy, 1 µM wortmannin (InvivoGen, San Diego, USA) or 1 µM SAR405 (Hycultec, Beutelsbach, Germany), and to inhibit ubiquitination, 1 µM ziram (Sigma-Aldrich) was added to neurons 16 hr before fixation.

## Western blot analyses

Cultured hippocampal neurons, either infected with lentivirus at 1 DIV or uninfected (control), were grown on 6-well-plates with a density of 25 k per 1 cm$^2$ until 13–15 DIV. All following steps were performed at 4°C. Neurons were kept on ice and washed twice with cold PBS. Subsequently, cells were detached by mechanical force. Isolated cells were centrifuged at 4000 rpm for 10 min and resuspended in 100 µl lysis buffer (50 mM Tris pH 7.9, 150 mM NaCl, 5 mM EDTA, 1% Triton X-100, 1% NP-40, 0.5% Deoxycholate, protease inhibitor cOmplete Tablets 1x) and incubated for 5 min on ice. Afterwards, cell suspension was centrifuged at 13,000 rpm for 10 min after which the supernatant was transferred into a new tube. Subsequently, the protein concentration was determined using the Pierce BCA Protein Assay Kit (Thermo Fisher Scientific). Synaptosomes were prepared as described previously (*Smalla et al., 2013*). In brief, brains were homogenized in homogenization buffer (5 mM HEPES pH 7.4, 320 mM sucrose and protease inhibitor cOmplete Tablets 1x), centrifuged at 1,000 g for 10 min and the pellet re-homogenized before an additional 10 min at 1,000 g. The supernatants were centrifuged at 12,000 g for 20 min. Subsequently, the pellet was re-homogenized in homogenization buffer and centrifuged at 12,000 g for 20 min resulting in the membrane fraction P2. Afterwards, the P2 fraction in 5 mM Tris/HCl pH 8.1 containing 320 mM sucrose was loaded on a sucrose density step gradient (0.85M, 1.0M, 1.2M). Subcellular fractionation was achieved by centrifugation at 85,000 g for 2 hr. Synaptosomes were harvested from the 1.0/1.2M interface. Next, protein concentrations were determined by BCA assay (Sigma-Aldrich), sucrose was removed by two centrifugation steps at 100,000 g for 30 min each and the resulting pellet was solubilized in an appropriate volume of SDS-PAGE loading buffer.

The same amount of total protein was then separated by SDS-PAGE and transferred onto a PVDF or nitrocellulose membrane. Afterwards, the membrane was blocked in 5% milk powder in tris buffered saline (TBS) with Tween-20 (TBS-T) (20 mM Tris, 150 mM NaCl, 0.1% Tween-20) for 1 hr followed by primary antibody incubation (1:1000 in 3% milk powder in TBS-T) over night at 4°C. The following antibodies were used: primary antibodies against Parkin (1:500; mouse; abcam, Cambridge, UK; Cat# ab77924), Siah1 (1:500; rabbit; abcam; Cat# ab203198), LC3 (1:1000; rabbit; Cell Signaling Technology, Cambridge, UK; Cat# 12741S), GAPDH (1:1000; rabbit; Cell Signaling Technology; Cat# 2118S) and Actin (1:1000; rabbit; Sigma-Aldrich; Cat# A2066). Afterwards, the membrane was washed three times with TBS-T for 10 min each and incubated with the secondary antibody (1:2500 in 3% milk powder in TBS-T) for 1 hr at RT. Horse radish peroxidase (HRP)-conjugated secondary antibodies were diluted 1:25000 (Sigma-Aldrich). Afterwards, the membrane was washed three times with TBS-T and bands were visualized using 20x LumiGLO Reagent and 20x Peroxidase (Cell Signaling Technology).

## Synaptotagmin1 uptake experiments

Primary hippocampal neurons from *Bsn* KO and WT mice (uninfected or infected with FU-shParkin) were washed two times with Tyrodes buffer (119 mM NaCl, 2.5 mM KCl, 2 mM CaCl$_2$, 2 mM MgCl$_2$, 30 mM glucose, 25 mM HEPES, pH 7.4) and then incubated with Oyster 550-labeled Synaptotagmin1 antibody (1:70; rabbit; Synaptic Systems; Cat# 105103C3) in Tyrodes buffer for 20 min at 37°C. Afterwards, cells were washed twice with Tyrodes buffer and subsequently fixed in 4% PFA in PBS. To inhibit autophagy, 1 µM wortmannin (InvivoGen) was added to neurons 16 hr before the assay. To quantify total Synaptotagmin1 levels, cells were stained post-hoc with antibodies against Synaptotagmin1 as described in *Immunocytochemistry of hippocampal neurons*.

## Electron microscopy

Cultured hippocampal neurons from *Bsn* KO and WT were plated on astrocytes grown on 6 mm sapphire disks at a density of 50 k per 1 cm$^2$. After a total of 14–17 days in culture, 1 µM wortmannin (InvivoGen) was added to specific samples 16 hr before cryo-fixation to inhibit autophagy. Then, all sapphire disks were cryo-fixed using a high pressure freezing machine (EM-ICE, Leica, Wetzlar,

Germany) in Base$^+$-solution containing the following: 140 mM NaCl, 2.4 mM KCl, 10 mM HEPES, 10 mM glucose, 2 mM CaCl$_2$, and 4 mM MgCl$_2$, pH7.4 (~300 mOsm). After freezing, samples were cryo-substituted in anhydrous acetone containing 1% glutaraldehyde, 1% osmium tetroxide and 1% milliQ water in an automated freeze-substitution device (AFS2, Leica). The temperature was kept for 4 hr at −90°C, brought to −20°C (5 °C/h), kept for 12 hr at −20°C and then brought from −20°C to +20°C. Once at room temperature, samples were *en-bloc* stained in 0.1% uranyl acetate, infiltrated in increasing concentration of Epoxy resin (Epon 812, EMS Adhesives, Delaware, USA) in acetone and finally embedded in Epon for 48 hr at 65°C. Sapphire disks were removed from the cured resin block by thermal shock. 50 nm thick sections were obtained using an Ultracut ultramicrotome (UCT, Leica) equipped with an Ultra 45 diamond knife (Ultra 45, DiATOME, Hatfield, USA) and collected on formvar-coated 200-mesh copper grids (EMS). Sections were counterstained with uranyl acetate and lead citrate and imaged in a FEI Tecnai G20 Transmission Electron Microscope (FEI, Hillsboro, USA) operated at 80–200 keV and equipped with a Veleta 2K × 2K CCD camera (Olympus, Hamburg, Germany). Around 150 electron micrographs were collected (pixel size = 0.7 nm) for each sample from at least two different experiments. Data were analyzed blindly using the ImageJ software. Double-membraned structures (autophagic vacuoles; AVs), single-membraned structures with vesicles inside (multivesicular bodies; MVBs) and synaptic vesicles (SVs) per synaptic junction were counted.

## Mass spectrometry

Synaptic protein extraction was performed using Syn-PER Synaptic Protein Extraction Reagent (Thermo Fisher Scientific) according to the manufacturer's instructions. Briefly, frozen cortices from 15-week-old WT and *Bsn* KO animals were diluted in 1 ml Syn-PER per 100 µg of tissue. Dounce homogenization was performed on ice using 10 slow strokes and the homogenate was transferred into a fresh tube. Centrifugation was performed twice at 1,200 g for 10 min at 4°C and the supernatant transferred into a fresh tube. Next, two centrifugation steps of 15,000 g for 20 min at 4°C resulted in a pellet which was resuspended in 200 µl urea buffer (6M Urea (Bio-Rad, Hercules, USA), 2M Thiourea (GE Healthcare, Chicago, USA), 10 mM HEPES (Carl Roth, Karlsruhe, Germany), pH 8). Protein concentration was determined using Pierce 660 nm Protein Assay Kit (Thermo Fisher Scientific). Samples were adjusted to 2,2 mg per 200 µl urea buffer. Disulfide bonds were reduced and alkylated by first adding Dithiothreitol (DTT, AppliChem, Darmstadt, Germany) with a final concentration of 6 mM in ABC (ammonium bicarbonate, Sigma-Aldrich) buffer (50 mM) for 30 min at RT. 2-chloracetamide (Honeywell Fluka, Charlotte, USA) was added with a final concentration of 10 mM in ABC buffer for 20 min at RT in the dark. 1 µg Lys-C (Wako Chemicals, Neuss, Germany) was added to 50 µg protein for 4 hr at RT. Subsequently, samples were diluted in 50 mM ABC buffer (1:4) and 1 µg Trypsin (Promega, Madison, USA) was added per 50 µg protein. Samples were digested overnight at RT in a ThermoMixer (Eppendorf, Hamburg, Germany). Trypsin digestion was stopped by acidifying the samples with trifluoracetic acid (TFA) (VWR, Radnor, USA) with a final concentration of 0.5%. For solid-phase extraction, SEP-Pak Vac 1cc (50 mg) C18 cartridges (WAT054955, Waters, Milford, USA) were used. Cartridges were washed with 600 µl of 70% Acetonitril (ACN) (Thermo Fisher Scientific) and equilibrated with 600 µl of 0.1% TFA. Samples were loaded and washed with 600 µl of 0.1% TFA and then eluted using 400 µl of 50% ACN/0.1% TFA and 400 µl of 70% ACN/0.1% TFA. The eluates were collected into fresh tubes and frozen at −80°C for at least 2 hr and subsequently lyophilized. Immunoaffinity purification was performed with PTMScan Ubiquitin Remnant Motif (K-ε-GG) Kit (#5562, Cell Signaling Technology) according to the manufacturer's instructions with the following adjustments: After washing with PBS, antibody beads (280 µl) were split into 7 equal parts of 40 µl each and transferred into fresh tubes for sample binding for 3 hr at 4°C. Afterwards, samples were loaded twice onto the SEP-Pak Vac 1cc (50 mg) C18 cartridges. After the purification procedure, samples were eluted into fresh tubes and frozen at −80°C overnight. Samples were lyophilized for 48 hr.

Enriched samples were analyzed by online nanoflow liquid chromatography tandem mass spectrometry (LC–MS/MS) on a Q Exactive Plus Quadrupole-Orbitrap mass spectrometer. Briefly, nano LC–MS/MS-experiments were performed on an Ultimate 3000RSC LC system connected to a Q Exactive Plus (Thermo Fisher Scientific) through a nanoelectrospray ion source. The enriched peptides were trapped on a precolumn (PepMap C18, 5 mm x 300 µm x 5 µm, 100Å, Thermo Fisher Scientific) with 2:98 (v/v) acetonitrile/water containing 0.1% (v/v) trifluoroacetic acid at a flow rate of 20

µl/min for 3 min and then separated by a 250 mm nano LC column (Acclaim PepMap C18, 2 µm; 100 Å; 75 µm, Thermo Fisher Scientific). The mobile phase (A) was 0.1% (v/v) formic acid in water, and (B) 80% acetonitrile in water containing 0.1% (v/v) formic acid. A gradient 3–40% (B) in 120′ with a flow rate of 300 nl / min was used. Total acquisition time was 155 min. The Q Exactive Plus instrument was operated in the data dependent mode to automatically switch between full scan MS and MS/MS acquisition. Survey full scan MS spectra (*m/z* 400–1650) were acquired in the Orbitrap with 70,000 resolution after accumulation of ions to a $3 \times 10^6$ target value based on predictive AGC from the previous full scan. Dynamic exclusion was set to 20 s. The eight most intense multiply charged ions ($z \geq 2$) were sequentially isolated and fragmented in the octopole collision cell by higher-energy collisional dissociation (HCD) with a fixed injection time of 200 ms and 17,500 resolution. The MS/MS ion selection threshold was set to $1 \times 10^5$ counts.

Proteins and peptides were identified by Mascot software version 2.6.1. (Matrix Science Ltd., London, UK) using the following search parameter set: SwissProt database, taxonomy *Mus musculus* enzyme: trypsin/P with three missed cleavages, static modification: Carbamidomethylation (C), variable modifications: Deamidation (NQ); GlyGly (K); Oxidation (M) and Acetyl (Protein N-term), mass tolerances for MS and MS/MS: 5 ppm and 0.02 Da, a calculated peptide FDR 1%, and identification threshold of p<0.001 (unambiguous) or >0.001 (ambiguous) was set.

Relative label-free quantification was performed with MaxQuant software version 1.6.0.1 (*Cox et al., 2014*) and default Andromeda LFQ parameter. Spectra were matched to a *Mus musculus* database (17,016 reviewed entries, downloaded from uniprot.org), a contaminant, and decoy database. Search parameters as described above were used. The logarithm to the base 2 from the average of three animals per genotype (0.00 were excluded before averaging) was plotted.

## Image acquisition and quantification

Images were acquired on a spinning disc confocal microscope (Zeiss Axio Oberserver.Z1 with Andor spinning disc and cobolt, omricron, i-beam laser) (Zeiss, Oberkochen, Germany) using either a 40x or $63 \times 1.4$ NA Plan-Apochromat oil objective and an iXon ultra (Andor, Belfast, UK) camera controlled by iQ software (Andor). Or images were acquired on a Zeiss Axio Imager A2 microscope with Cool Snap EZ camera (Visitron Systems GmbH, Puchheim, Germany) controlled by VisiView (Visitron Systems GmbH) software. Or images were acquired on an Olympus microscope IX83 (Olympus) with a Zyla camera (Andor) controlled by iQ software (Andor). Approximately 10 images of each sample per culture were processed using ImageJ and OpenView software (written by Dr. Noam Ziv, Technion Institute, Haifa, Israel) (*Tsuriel et al., 2006*). In brief with the OpenView software, multichannel intensities were measured using a semi-automatically box routine associated with individual boutons. Boxes were about 0.8 µm x 0.8 µm in size, whereas settings were kept the same (e.g. thresholds). The average fluorescence intensity was calculated from all quantified puncta per image. Afterwards, the average fluorescence intensities per image were normalized to the control group (WT) for each experiment. For quantification of number of puncta, approximately 10 images of each sample per culture were taken. From each image, 1–2 separated axonal segments, defined by the thin and smooth (i.e. protrusion-free) morphology and the alignment with Synaptophysin1 puncta present on uninfected neurons, were randomly picked and the number of puncta per unit length was counted manually.

## Experimental Design and statistical Analyses

Statistical design for all experiments can be found in the figure legends. Independent experiments equal independent cultures. For western blot and mass spectrometry analyses, 3–4 animals per genotype were used. All data were included in the final quantification, except for electron microscopy analyses, where 3 outliers for SVs/µm² were excluded (446.81 in Bsn KO; 596.19 in WT + wtm; 539.44 in Bsn KO + wtm). All data representations and statistical analyses were performed with Graph-pad Prism.

## Acknowledgements

We thank Prof. Richard J Reimer and Prof. Noam E Ziv for discussion and valuable comments on the manuscript, Beatriz Villafranca Magdalena, Anny Kretschmer, Katja Czieselsky, Bettina Brokowski, Katja Poetschke, Janina Juhle, Kathrin Pohlmann, Peggy Patella and the animal facility at LIN

Magdeburg for technical assistance, the Virus Core Facility of the Charité - Universitätsmedizin Berlin for virus production and the Core Facility High Throughput Mass Spectrometry of the Charité for mass spectrometry.

## Additional information

### Funding

| Funder | Grant reference number | Author |
|---|---|---|
| Federal Government of Germany | SFB958 | Craig Curtis Garner |
| Federal Government of Germany | SFB779/B09 | Eckart D Gundelfinger |
| BMBF | 20150065 | Karl-Heinz Smalla<br>Eckart D Gundelfinger |

The funders had no role in study design, data collection and interpretation, or the decision to submit the work for publication.

### Author contributions

Sheila Hoffmann-Conaway, Carolina Montenegro-Venegas, Conceptualization, Data curation, Formal analysis, Investigation, Writing - original draft, Writing - review and editing; Marisa M Brockmann, Katharina Schneider, Anil Annamneedi, Kazi Atikur Rahman, Christine Bruns, Kathrin Textoris-Taube, Data curation, Formal analysis; Thorsten Trimbuch, Viral construct preparation; Karl-Heinz Smalla, Christian Rosenmund, Conceptualization, Resources; Eckart D Gundelfinger, Craig Curtis Garner, Conceptualization, Funding acquisition, Writing - review and editing

### Author ORCIDs

Marisa M Brockmann (iD) http://orcid.org/0000-0002-1386-5359
Kazi Atikur Rahman (iD) http://orcid.org/0000-0001-8124-6026
Christian Rosenmund (iD) http://orcid.org/0000-0002-3905-2444
Eckart D Gundelfinger (iD) https://orcid.org/0000-0001-9377-7414
Craig Curtis Garner (iD) https://orcid.org/0000-0003-1970-5417

### Ethics

Animal experimentation: Breeding of animals and experiments using animal material were carried out in accordance with the European Communities Council Directive (2010/63/EU) and approved by the local animal care committees of Sachsen-Anhalt or the animal welfare committee of Charité Medical University and the Berlin state government (protocol number: T0036/14, O0208/16).

### Decision letter and Author response

Decision letter https://doi.org/10.7554/eLife.56590.sa1
Author response https://doi.org/10.7554/eLife.56590.sa2

## Additional files

### Supplementary files

• Transparent reporting form

### Data availability

All data generated or analysed during this study are included in the manuscript.

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
