## [Decision Letter]

**Acceptance summary:**

This study finds that Bassoon, a presynaptic active zone protein, inhibits the clearance of presynaptic vesicle proteins through the ubiquitin and autophagy pathways. In addition, the authors also find that the autophagy-mediated ubiquitination and clearance of synaptic vesicle proteins is critically mediated by the E3 ligase Parkin.

**Decision letter after peer review:**

Thank you for submitting your article "Parkin contributes to synaptic vesicle autophagy in Bassoon-deficient mice" for consideration by *eLife*. Your article has been reviewed by three peer reviewers, and the evaluation has been overseen by a Reviewing Editor and Richard Aldrich as the Senior Editor. The following individual involved in review of your submission has agreed to reveal their identity: Volker Haucke (Reviewer #3).

The reviewers have discussed the reviews with one another and the Reviewing Editor has drafted this decision to help you prepare a revised submission.

In revising the manuscript, please read the following review comments carefully and address all of them, except for those requiring additional experiments, either by toning down the main conclusions, performing additional analyses of existing datasets, or clarifying the text and presentation. In addition, Reviewer #3 suggested for you to check your existing EM images to see if there are any changes in mitochondrial patterns in Bassoon knockout mice. Please let us know if you have any questions in revising the manuscript.

Reviewer #1:

The study reports the key role of Bassoon, a presynaptic active zone protein, in the regulation of presynaptic autophagy and presynaptic vesicle (SV) proteostasis. The novelty of this study lies in that they found that autophagic, but not proteasomal or endo-lysosomal, pathway critically regulates SV proteins. Another point of novelty is that they found Parkin, among the ~six hundred known E3 ligases including Siah1, is a critical mediator of Bassoon deletion-dependent autophagic removal of SV proteins. The findings are again quite novel, and the datasets are in general of high quality.

1) Parkin knockdown and related results constitute a critical part of the main conclusions. The Parkin knockdown construct decreases the levels of Parkin in cultured neurons by ~70% (Figure 7—figure supplement 1). However, it is unclear whether this knockdown construct does not have off-target effects. This could be addressed by the rescue of the knockdown effects by an shRNA-resistant Parkin expression construct. The rescue effects could be monitored using either RFP-LC3 puncta or levels of SV proteins.

Reviewer #2:

This work addresses an important question: what are the mechanisms, at synapses, that degrade proteins to maintain function? Specifically, it examines the role of the synaptic protein Bassoon in its regulation of the autophagy pathway. The authors build on the findings from their two previous papers (Waites, 2013 and Okerlund, 2017) in which they established that loss of Bassoon promotes autophagy. In this manuscript the authors extend these findings, demonstrating Bassoon-dependent degradation of synaptic vesicles through the E3 ubiquitin ligase Parkin (and to a lesser extent Siah1). The manuscript addresses important questions, and we make suggestions below to strengthen the conclusions in the context of the findings.

1) The authors claim that Bassoon negatively regulates synaptic autophagy through its ability to inhibit enzymes involved in ubiquitination. But could defects in ubiquitination of the synaptic machinery also result from general problems in the structure of the synapse due to losing of Bassoon? For example, how specific are the reported ubiquitination defects in Bassoon as compared to Piccolo?

2) Along the same lines, can the authors discuss how they envision Bassoon regulates Parkin? E.g. is parkin expression level increased in Bassoon KO? If the level of Parkin is not increased, are known substrates of Parkin increased? I don't think these analyses will require additional experiments, just examination of their data and discussion.

3) Could the authors report the effect size for the quantifications?

4) It is hard to deconvolve the effects of reduced co-localization with a general reduction on the expression of the proteins. In that sense the conclusions for co-localization are confusing for some of the variables (like KO + Bsn609 in Figure 1E). If co-localization is to be analyzed, and a conclusion drawn, it would be good to control for the reduced fluorescence, and do analyses/plots similar to those in Okerlund, 2017.

5) In Figure 2D, adding UbK_0_ does not increase Syp level, does that indicate the Bsn609 may also induce other degradative pathways other than ubiquitination to break down Syp?

6) In Figure 4C, by live labeling Synaptotagmin1 antibody, the author intended to show the synaptic vesicle cycling in Bassoon KO can be rescued by blocking autophagy. How did they know if the Bassoon KO does not affect Syt1 degradation?

7) EM data is consistent with a loss of SVs. FM dyes showing less synaptic vesicle or less material internalization may be useful. Or authors can normalize 'uptaken' syt-1 to total syt-1, if those quantifications are available.These results are explained well in the Discussion, but authors should similarly explain the conclusions and the limitations of the experiment in the Results.

8) In Figure 5A, the text reads "AVs filled with SV-sized vesicles were readily observed in Bsn KO samples", but we had a hard time finding the SV-sized vesicles in the images. Zoomed in images with notations are recommended.

9) Are the differences in Figure 6C significant? Is there a positive control for it/baseline?

10) In Figure 6, did the authors examine WT and KO + wortmannin (or Bsn-ZnF overexpression, or by inhibiting poly-ubiquitinylation) to examine if this effect is specific to Bassoon-regulated autophagy?

11) Figures 7 and 8, Parkin instead of Siah1 is the major E3 ligase in Bassoon KO. Yet Bsn609 rescues the Bassoon phenotype and interacts with Siah1, and Bassoon did not interact with Parkin. Can the authors discuss this?Or Include a diagram of the model of Bassoon, Siah1, Parkin, and autophagy?

12) Knockdown efficiency for siah-1 is lower than that for Parkin. Were the effects normalized to the observed knockdown levels, i.e, how can authors rule out the possibility that the bigger effect by Parkin is due to better KD efficiency?

Reviewer #3:

In their manuscript Hoffmann-Conaway et al. follow-up from recent studies by the same group of authors (Okerlund et al., 2017; Hoffmann et al., 2019) on the role of the giant active zone protein Bassoon in the regulation of presynaptic autophagy and synaptic vesicle turnover. The authors compellingly demonstrate that Bassoon loss triggers the ubiquitination of a variety of presynaptic proteins including a subset of SV proteins such as SV2B and VAMP/ Synaptobrevin 2. Further studies by light and electron microscopy as well as lentiviral depletion of candidate E3 ligases suggests a role for Bassoon in restricting (via an unknown mechanism) the activity of the E3 ligase Parkin, which, when hyperactivated in Bassoon KO neurons triggers the ubiquitination and autophagic turnover of synaptic vesicles. Consistent with this they find elevated numbers of autophagic structures containing SV-like entities in Bassoon KO synapses, a phenotype that is repressed by the lipid kinase inhibitor wortmannin.

This is a well-written manuscript on an important, yet, under-researched topic in synaptic physiology that should be of broad interest to the neuroscience and cell biology communities. Most of the experiments are carefully designed and executed and support the authors' conclusions, although the mechanisms underlying the observed phenotypes remain to be determined. I have a number of points that need to be considered prior to publication of this manuscript in *eLife*.

1) Wortmannin is a rather non-specific irreversible PI kinase inhibitor that targets all PI 3-kinases and some PI 4-kinases. I suggest to repeat some of the key experiments using available specific Vps34 inhibitors (e.g. SAR-405, VPS34-IN1, Compound 19) or, alternatively, block autophagy by KD of key components such as ATG5, ATG7, or Vps34.

2) The authors propose a model based on their EM data according to which Bassoon-regulated Parkin-triggered autophagy targets entire SVs for lysosomal turnover. If this was the case the majority of SV proteins should display similar changes in their steady-state levels and half-life. I suggest that the authors examine SV proteins other than SV2B and VAMP/ Synaptobrevin 2 in Figure 4, e.g. SV2A, Syt1, vGLUT1, Synaptophysin. Do these behave similarly and, if so, are their levels rescued by autophagy inhibition in the presence of a specific Vps34 inhibitor (see above)? What happens to the steady-state levels of the many non-SV proteins detected in the mass spec analysis of ubiquitinated peptides, for example SNAP-25 and Syntaxin 1?

3) A large body of literature has implicated Parkin in mitophagy. What are the effects of Bassoon loss and concomitant inhibition of either Parkin or Vps34 on the levels of synaptic mitochondria detectable by light and electron microscopy?

4) Statistics: n appears to be axons rather than experiments. Some explanations are needed why this is justified and how axons are distinguished from dendrites in these cultures.

---

## [Author Response]

Reviewer #1:[…] 1) Parkin knockdown and related results constitute a critical part of the main conclusions. The Parkin knockdown construct decreases the levels of Parkin in cultured neurons by ~70% (Figure 7—figure supplement 1). However, it is unclear whether this knockdown construct does not have off-target effects. This could be addressed by the rescue of the knockdown effects by an shRNA-resistant Parkin expression construct. The rescue effects could be monitored using either RFP-LC3 puncta or levels of SV proteins.

We agree that using an shRNA-resistant Parkin as a control is an excellent idea and would have improved our manuscript. However, we did not do these experiments and cannot do them in the near future due to the current Covid-19 situation.

Reviewer #2:[…] The manuscript addresses important questions, and we make suggestions below to strengthen the conclusions in the context of the findings.1) The authors claim that Bassoon negatively regulates synaptic autophagy through its ability to inhibit enzymes involved in ubiquitination. But could defects in ubiquitination of the synaptic machinery also result from general problems in the structure of the synapse due to losing of Bassoon? For example, how specific are the reported ubiquitination defects in Bassoon as compared to Piccolo?

This is an interesting idea. On the one hand, we could not see any profound structural changes in our electron micrographs (Figure 5) suggesting that the overall synaptic ultra-structure is unaltered. This is in line with previous findings that normal basic electrophysiological measurements are unchanged in Bsn-deficient neurons (Altrock et al., 2003). On the other hand, Bsn KO animals are epileptic and changes in plasticity and in molecular organization were observed upon strong stimulation (e.g. Hallermann et al., 2010; for review see: Gundelfinger et al., fnsyn.2015.00019). Binding to Atg-5 argues for a direct involvement of Bassoon in autophagic regulation (Okerlund et al., 2017). But, indeed, the mode how Parkin is activated in the absence of Bassoon remains unclear. Concerning Piccolo-deficiency much milder phenotypes were observed in Piccolo-mutant mice (Mukherjee et al., 2010). More over our analysis of Piccolo KO rats, reveal no significant changes in the structure of the active zone, though defects in SV recycling were detected (Ackermann et al., 2019). The idea of analyzing the ubiquitination levels in these KO animals is a great one, which we plan to pursue in the future. Unfortunately, at present, we cannot make any assumptions on the specificity of ubiquitination in loss of Bassoon versus loss of Piccolo. Nonetheless, to address the justified comment of the reviewer, we have modified our wording concerning autophagy triggered by the absence of Bassoon throughout the manuscript.

2) Along the same lines, can the authors discuss how they envision Bassoon regulates Parkin? E.g. is parkin expression level increased in Bassoon KO? If the level of Parkin is not increased, are known substrates of Parkin increased? I don't think these analyses will require additional experiments, just examination of their data and discussion.

We agree that in future studies it would be worthwhile investigating the exact mechanisms by which Bassoon can regulate Parkin activity. Unfortunately, in western blot analyses we could not find any differences in Parkin expression levels between synaptosomes or homogenates of Bsn KO mice compared to WT animals, nor could we co-immuno-precipitate these two potential partners. This leads us to envision Bassoon and Parkin as parts of a larger assembly complex, or that the absence of Bassoon, induces an alternative Parkin-activating pathway (Discussion, eleventh paragraph). Intriguingly, one described substrate of Parkin, Synaptotagmin11 (Discussion, tenth paragraph), was highly ubiquitinated in Bsn KO neurons, further indicating that the loss of Bassoon could lead to an over-activation of Parkin. This latter concept is supported by the rescue of Bassoon related autophagy phenotypes in neurons with reduced levels of Parkin (Figure 9).

3) Could the authors report the effect size for the quantifications?

Actually the effect sizes are given in all relevant figure legends.

4) It is hard to deconvolve the effects of reduced co-localization with a general reduction on the expression of the proteins. In that sense the conclusions for co-localization are confusing for some of the variables (like KO + Bsn609 in Figure 1E). If co-localization is to be analyzed, and a conclusion drawn, it would be good to control for the reduced fluorescence, and do analyses/plots similar to those in Okerlund, 2017.

Thank you for reminding us of this work. Actually, in the Okerlund et al., 2017 study, the quantification of colocalization was performed as follows: “Synaptic puncta (SV2, synapsin1a or synaptophysin) and autophagic puncta (LC3 or Atg5) were counted, and the number of overlapping puncta between the synaptic and autophagic puncta were counted and calculated as ratio of colocalized puncta to synaptic puncta.” In the current study, we used the same approach. However, we strongly agree that line scans, as shown in Okerlund et al., 2017, can help to visualize colocalization. Therefore, we prepared a new supplementary figure (Figure 1—figure supplement 1) showing line scans for all analyzed colocalizations.

5) In Figure 2D, adding UbK_0_ does not increase Syp level, does that indicate the Bsn609 may also induce other degradative pathways other than ubiquitination to break down Syp?

At this point, we do not know why Bsn609 increases Synaptophysin levels, while the UbK_0_ does not. One possibility is the ubiquitination-independent degradation of Synaptophysin. On the other hand, the N-terminal fragment of Bsn609 associates with SVs and accumulates in presynaptic boutons (Dresbach et al., 2003), which might have an additional effect on the local clearance rates of SV proteins. How this is mediated mechanistically is unclear and would require additional experiments beyond the Covid-19 constraints.

6) In Figure 4C, by live labeling Synaptotagmin1 antibody, the author intended to show the synaptic vesicle cycling in Bassoon KO can be rescued by blocking autophagy. How did they know if the Bassoon KO does not affect Syt1 degradation?

We agree that lower levels of Synaptotagmin1 per se could also lead to a decrease in Synaptotagmin1 antibody uptake. Fortunately, we were able to analyze already performed experiments that are also related to query 7. In particular, Synaptotagmin1 uptake cultures were post-hoc stained with antibodies against Synaptotagmin1 to visualize total Synaptotagmin1 pools. Interestingly, while Synaptotagmin1 antibody uptake was significantly decreased in Bsn KO neurons compared to WT neurons, total Synaptotagmin1 levels were not (new supplementary Figure 4—figure supplement 2). This observation leads us to the assumption that Synaptotagmin1 antibody uptake can be used as a measure of SV cycling in these animals.

7) EM data is consistent with a loss of SVs. FM dyes showing less synaptic vesicle or less material internalization may be useful. Or authors can normalize 'uptaken' syt-1 to total syt-1, if those quantifications are available.These results are explained well in the Discussion, but authors should similarly explain the conclusions and the limitations of the experiment in the Results.

This is a very interesting idea related to query 6. As described above, we were able to add new supplementary data (from already performed experiments) showing not only that total Synaptotagmin1 levels were unchanged, but also that normalized Synaptotagmin1 uptake was significantly decreased in Bsn KO neurons (Figure 4—figure supplement 2). Using FM dyes to verify Synaptotagmin1 antibody uptake results is an excellent idea, which we will definitely consider for future studies after the Covid-19 situation is resolved.

8) In Figure 5A, the text reads "AVs filled with SV-sized vesicles were readily observed in Bsn KO samples", but we had a hard time finding the SV-sized vesicles in the images. Zoomed in images with notations are recommended.

To increase visibility of the structures in EM, we increased the size of the images by 15% and added a zoom (Figure 5A) of the double-membraned structure that appears to have engulfed vesicles with diameters of approx. 50nm.

9) Are the differences in Figure 6C significant? Is there a positive control for it/baseline?

To clarify, the histogram in Figure 6C is only an additional way of illustrating the data shown in Figure 6B, where a t test was used to determine significant changes between WT and Bsn KO (p***=0.0008). We modified the text to express that more accurately.

As the images were taken from fixed samples, the ratio of old/young protein in WT synapses functions as a control. However, in initial control experiments, we did check that timer molecules function robustly. Here, we treated WT cells with cycloheximide, a protein synthesis inhibitor, which led to an overall older pool of SV2 (data not shown).

10) In Figure 6, did the authors examine WT and KO + wortmannin (or Bsn-ZnF overexpression, or by inhibiting poly-ubiquitinylation) to examine if this effect is specific to Bassoon-regulated autophagy?

Yes, we were starting to look into these important questions. However, we were not able to acquire enough data to verify these results, due to the current Covid-19 situation and the subsequent restricted access to our labs. Future studies addressing these will be very interesting.

11) Figures 7 and 8, Parkin instead of Siah1 is the major E3 ligase in Bassoon KO. Yet Bsn609 rescues the Bassoon phenotype and interacts with Siah1, and Bassoon did not interact with Parkin. Can the authors discuss this?Or Include a diagram of the model of Bassoon, Siah1, Parkin, and autophagy?

This is an important question. Specifically, as the reviewer noted below, one could explain the weaker impact of Siah1 KD versus Parkin KD as being simply due to a better KD efficiency. Further hints that Siah1 is perhaps yet more important are suggested by the data with Bsn609, which we previously have shown to bind and efficiently inhibit Siah1 (Waites et al., 2013). As Bsn609 does not appear to bind Parkin, and yet Parkin KD efficiently rescues, we anticipate that either a different region of Bassoon binds and regulates the activity of Parkin, or Bassoon acts to regulate Parkin via an indirect mechanisms. Clearly, sorting out these important issues will require deeper evaluation, something we would endeavor to explore after the Covid-19 situation is resolved.

12) Knockdown efficiency for siah-1 is lower than that for Parkin. Were the effects normalized to the observed knockdown levels, i.e., how can authors rule out the possibility that the bigger effect by Parkin is due to better KD efficiency?

This is an important question and related to query 11. Given the stronger effect of Bsn609 and its ability to bind and inhibit the activity of Siah1, we would agree with the reviewer’s suggestion that the bigger effect by KD of Parkin is due to a higher KD efficiency. We adjusted the Discussion accordingly.

Reviewer #3:[…] I have a number of points that need to be considered prior to publication of this manuscript in eLife.1) Wortmannin is a rather non-specific irreversible PI kinase inhibitor that targets all PI 3-kinases and some PI 4-kinases. I suggest to repeat some of the key experiments using available specific Vps34 inhibitors (e.g. SAR-405, VPS34-IN1, Compound 19) or, alternatively, block autophagy by KD of key components such as ATG5, ATG7, or Vps34.

This is an excellent suggestion. We already performed a few control experiments quantifying RFP-LC3 puncta and SV2b intensities, with and without the specific Vps34 inhibitor SAR405. We made a small supplementary figure to strengthen our wortmannin results (Figure 4—figure supplement 1).

2) The authors propose a model based on their EM data according to which Bassoon-regulated Parkin-triggered autophagy targets entire SVs for lysosomal turnover. If this was the case the majority of SV proteins should display similar changes in their steady-state levels and half-life. I suggest that the authors examine SV proteins other than SV2B and VAMP/ Synaptobrevin 2 in Figure 4, e.g. SV2A, Syt1, vGLUT1, Synaptophysin. Do these behave similarly and, if so, are their levels rescued by autophagy inhibition in the presence of a specific Vps34 inhibitor (see above)? What happens to the steady-state levels of the many non-SV proteins detected in the mass spec analysis of ubiquitinated peptides, for example SNAP-25 and Syntaxin 1?

We strongly agree that the analysis of other SV and non-SV proteins would have improved our manuscript. Unfortunately, we did not do the majority of these experiments and cannot do them in the near future due to lab lock-down in the current Covid-19 situation. Nevertheless, we were able to add a new supplementary figure (Figure 4—figure supplement 2) showing the analysis of already performed experiments. Interestingly, we observed no significant changes in Synaptotagmin1 levels between Bsn KO and WT neurons indicating that not all SV proteins are decreased in Bsn KO neurons. This finding demonstrates how important it is to investigate more proteins in the future. We plan on addressing these open questions once experimental work can be conducted normally again.

3) A large body of literature has implicated Parkin in mitophagy. What are the effects of Bassoon loss and concomitant inhibition of either Parkin or Vps34 on the levels of synaptic mitochondria detectable by light and electron microscopy?

This is a very interesting idea. We acknowledge that Parkin is not only involved in the ubiquitination of synaptic and SV proteins, but was predominantly described to be required for mitophagy. We have preliminary data indicating that the loss of Bsn causes a trend towards decreased numbers of mitochondria per terminal. We plan to follow-up on this in future studies.

4) Statistics: n appears to be axons rather than experiments. Some explanations are needed why this is justified and how axons are distinguished from dendrites in these cultures.

Yes, the depicted n number is either the number of axonal segments or the number of images that were used for analysis. It is common practice to show axons, cells or images as data points because these can be seen as the unit of variation. We added a more detailed description of our n numbers and how axons are distinguished from dendrites in the Materials and methods section. Another important feature of our analysis is that we pick RFP-LC3 expressing axons on top of uninfected neurons to avoid dendritic background. This way, and by strong colocalization of Synaptophysin1 along axons filled with soluble RFP (not conjugated to autophagosomes), we are certain to quantify puncta from axons only.